# Regulatory mechanisms underlying coordination of amino acid and glucose catabolism in *Escherichia coli*

Mattia Zampieri [1,2], Manuel Hörl[1,2], Florian Hotz[1], Nicola F. Müller[1] & Uwe Sauer[1]

How microbes dynamically coordinate uptake and simultaneous utilization of nutrients in complex nutritional ecosystems is still an open question. Here, we develop a constraint-based modeling approach that exploits non-targeted exo-metabolomics data to unravel adaptive decision-making processes in dynamic nutritional environments. We thereby investigate metabolic adaptation of *Escherichia coli* to continuously changing conditions during batch growth in complex medium. Unexpectedly, model-based analysis of time resolved exo-metabolome data revealed that fastest growth coincides with preferred catabolism of amino acids, which, in turn, reduces glucose uptake and increases acetate overflow. We show that high intracellular levels of the amino acid degradation metabolites pyruvate and oxaloacetate can directly inhibit the phosphotransferase system (PTS), and reveal their functional role in mediating regulatory decisions for uptake and catabolism of alternative carbon sources. Overall, the proposed methodology expands the spectrum of possible applications of flux balance analysis to decipher metabolic adaptation mechanisms in naturally occurring habitats and diverse organisms.

[1] Institute of Molecular Systems Biology, ETH Zürich, Zürich, 8093, Switzerland. [2] These authors contributed equally: Mattia Zampieri, Manuel Hörl. Correspondence and requests for materials should be addressed to M.Z. (email: zampieri@imsb.biol.ethz.ch) or to U.S. (email: sauer@imsb.biol.ethz.ch)

In natural environments, bacteria are frequently exposed to continuous changes in the flow of nutrients[1,2]. To ensure growth and survival, bacteria constantly sense metabolic changes and rapidly adapt to the appearance or depletion of nutrients[3]. To this end, cells integrate complex and dynamic external inputs to decide, which nutrients to consume and when to switch to consumption of alternative available substrates[4,5]. Monitoring dynamic changes in intracellular metabolism is key to understand regulatory mechanisms underlying adaptation to fluctuating environments[6,7]. However, resolving dynamic flux rearrangements is challenging, in particular in complex media with parallel consumption of multiple nutrients[8,9].

The most rigorous approaches infer intracellular fluxes from isotopic tracer experiments that are typically limited to steady state conditions and consumption of a single growth-limiting nutrient[10–12]. Alternatively, by assuming optimality functions of cellular behavior, fluxes can be estimated from constraint-based modeling of the metabolic network stoichiometry, such as in flux balance analysis (FBA)[13–15]. While FBA-like methods provide network-wide flux solutions, simplistic objective functions, such as maximization of growth, have been met with limited success for complex media conditions[16,17]. In such complex and dynamic environments without a defined limiting substrate, predicting metabolic fluxes using FBA remains a major challenge[18]. New analytical methodologies capable of tackling the chemical complexity of natural environments and of directly measuring rapid changes in the abundance of multiple substrates and byproducts in parallel, can provide additional constraints to model intracellular metabolism and open new opportunities to shed light on the mechanisms underlying decision-making processes of bacteria dealing with complex and dynamic nutritional conditions.

Here, we exploit non-targeted mass-spectrometry[19] and developed a novel computational approach to resolve dynamic metabolic adaptation of bacteria in complex chemical environments featuring continuous changes in available nutrients. Specifically, we applied this methodology to *Escherichia coli* growing in glucose minimal medium supplemented with casamino acids, an undefined mixture of amino acids and oligopeptides, as a proxy for undefined environmental conditions[20]. Continuous changes in nutrients availability are monitored by mass-spectrometry profiling of relative nutrient and by-product abundance in the supernatant. To meet the time-resolution and coverage demands for fluctuating environments with large chemical diversity, we monitored relative time-dependent changes in extracellular metabolite concentrations using a high-throughput non-targeted mass-spectrometry method[21,22]. Constraint-based modeling of metabolome dynamics reveals temporal coordination of glucose and amino acid catabolism, in which degradation of low-cost amino acids, into oxaloacetate and pyruvate, is responsible for a reduced glucose uptake and increased acetate secretion.

## Results

### Inference of flux dynamics from exo-metabolome profiles.
As an example of a complex nutritional environment, we cultivated *E. coli* wild-type *BW25113* in M9 minimal medium with glucose (5 g/L) and casamino acids (CAA) (2 g/L), an undefined mixture of amino acids and oligopeptides from digested milk protein[23]. The culture exhibited a classical batch growth curve with 4 h of growth fluctuating around maximum exponential growth rate, followed by a continuous decline in growth rate (Fig. 1a and Supplementary Fig. 1). To identify the nutritional conditions that underlie the changes in growth rate, we collected aliquots of culture supernatant at ten time points over the growth curve. To maximize the coverage of metabolites in the medium and to scale this approach to high-resolved time measurements of dynamic

changes in the exo-metabolome[24], we monitored the relative abundance of 8091 ions by flow-injection time-of-flight mass-spectrometry (FIA-TOFMS)[21]. 427 of these ions could be putatively annotated as deprotonated metabolites, which corresponded to nearly 40% of the metabolites represented in a genome-scale model of *E. coli*[25], and 61% when considering only metabolites with a known active or passive transport mechanism (Fig. 1b). One-hundred sixty-seven of the 427 detected metabolites were already present in the medium, demonstrating that CAA are a potentially complex source of alternative nutrients beyond amino acids. In particular, we detected several nucleotide precursors (Supplementary Fig. 2). To verify whether these compounds were present at relevant physiological concentrations, we showed that CAA can support the growth of *E. coli* gene knockouts that are auxotrophic for uridine, GMP, biotin, and xanthine (Supplementary Fig. 3). Of the 427 metabolites detected in the spent medium, 35% were only consumed and 34% secreted (Fig. 1c). The latter metabolites presumably result from overflow metabolism that goes beyond central metabolism[26,27], and involve peripheral pathways such as nucleotide metabolism, consistent with earlier observations[28] (Supplementary Fig. 4a). The remaining metabolites, such as intermediates of the tricarboxylic acid (TCA) cycle (Supplementary Fig. 4b), exhibited more complex secretion and consumption patterns.

While our metabolomics platform revealed unexpectedly complex exo-metabolome dynamics, suggesting widespread metabolite secretion as a homeostatic mechanism, the obtained relative concentration changes do not allow for a direct inference of intracellular fluxes within a classical FBA framework[8,15,29]. To integrate such non-targeted, direct flow-injection metabolomics measurements[21] in a genome-scale model for dynamic flux estimation, we used constraint-based modeling and developed a framework that consists of three main steps. First, we determined absolute concentrations for abundant medium components, namely glucose, acetate and 20 amino acids (Supplementary Data 1). Glucose and acetate were measured using existing enzymatic kits, and for amino acids we set-up calibration curves on our non-targeted mass-spectrometry platform (see Supplementary Methods). The quality of this non-targeted amino acid quantification is comparable to that of a standard High-Pressure Liquid Chromatography (HPLC) method[30] (Supplementary Fig. 5), but has a several-fold higher throughput and sensitivity. Second, the time-dependent profiles of all detected metabolites, including glucose, acetate, amino acids, and the remaining 407 metabolites detected by FIA-TOFMS (Supplementary Data 1), were interpolated using multivariate adaptive regression splines (MARS)[31]. MARS models are based on piece wise regression and automatically determine the number of basis spline functions and knot locations without the necessity for manual curation[31]. To estimate variability in fitting estimates due to noise in the measurements, we used a bootstrapping approach. For each metabolite, we repeated the fitting 1000 times using only 90% of the data. Error estimates were calculated as the standard deviation over the fittings obtained from downsampled data (e.g., gray shaded region in Fig. 2c). Next, we estimated relative instantaneous uptake and secretion rates by calculating the difference of metabolite levels between two consecutive time points, divided by the change in optical density ($OD_{600}$) and multiplied by the instantaneous growth rate (Supplementary Data 1). Third, we incorporated the estimated uptake and secretion rates in a constraint-based approach to resolve dynamic intracellular metabolic responses to changes in nutrient availability. Akin to other dynamic FBA modeling approaches[8,17,32], bacterial growth was divided into $N$ intervals at equidistant optical densities (ODs) and fluxes were assumed to be constant within these intervals. Absolute and relative consumption/secretion rates are used as soft

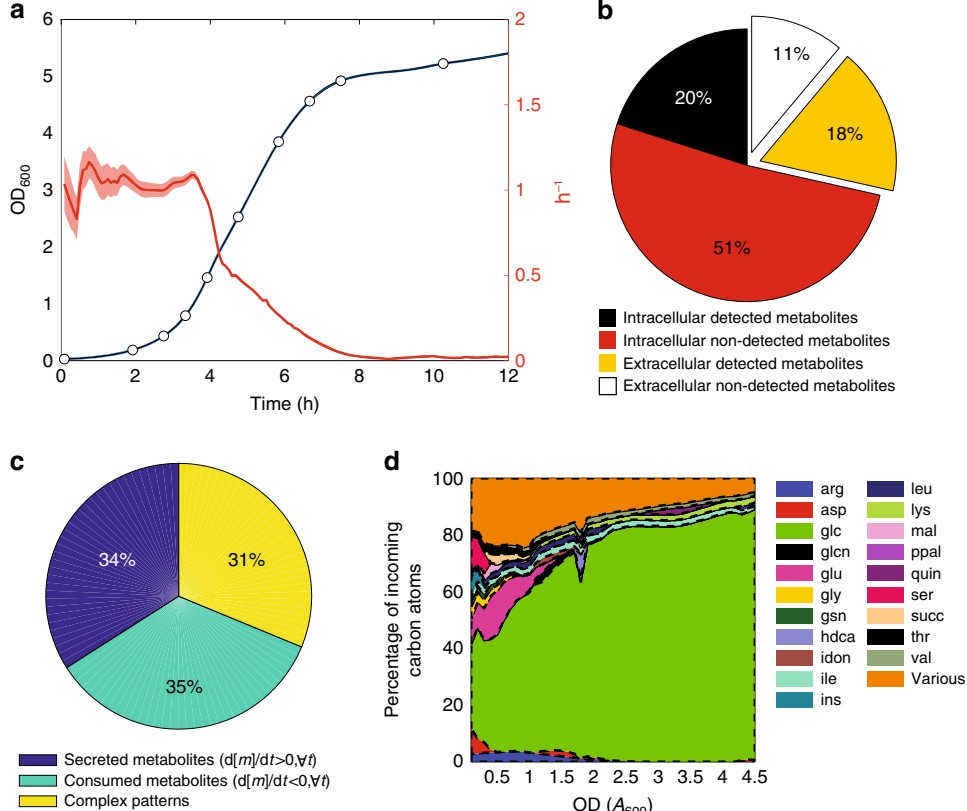

**Fig. 1** Exo-metabolome profiling. **a** Optical densities (OD$_{600}$; blue line) and instantaneous growth rate—i.e., OD$_{600}$ difference between two contiguous time points divided by the time interval (red line), of an *E. coli* culture growing in M9 glucose minimal medium supplemented with casamino acids (CAA). Dots represent time points for withdrawal of supernatant aliquots for exo-metabolome analysis. The solid line represents the mean across four biological replicates, while the shaded region is the mean ± standard deviation. **b** Percentage of detected metabolites with respect to the annotated compounds in the genome-scale model of *E. coli*[25]: metabolites that can be exchanged with the external environment (white and yellow), and those that are constrained to be intracellular (red and black). **c** Percentage of detected metabolites exhibiting complex (yellow) or monotonic profiles, i.e., instantaneous derivative in time is always negative (green—consumed) or positive (blue—secreted). **d** The total incoming flux of carbon at each given time is calculated by multiplying the number of carbon atoms for each compound and the respective estimated uptake flux. The relative percentage is reported. Only those metabolites contributing for at least 2% of the total incoming carbon are listed. The remaining compounds are grouped (i.e., Various in the legend). Correspondence between metabolite IDs in the legend and metabolite names can be found in the Supplementary Data 1

constraints to reduce the space of feasible flux solutions. To account for the variability in estimated rates, lower and upper bounds for each exchange flux (i.e., mean ± standard deviation determined by the bootstrapping approach) are included in the model. As changes in OD$_{600}$ can be routinely measured at high time-resolution and accuracy using standard plate readers, experimentally measured growth rates are used as hard constraint in the model. Hence, maximization of the biomass objective does not need to be invoked. For metabolites without external calibration, we introduce an auxiliary time-independent variable representing the proportional scaling factor (**c**) between measured MS intensities and actual concentrations. Instead of solving fluxes (**v**($t$)) at each time point independently[33], our method is formulated as a one-step global linear optimization problem to generate time dependent flux maps. The single constraint-based model contains genome-scale networks (*S*) for each time point, time-specific bounds on uptake and secretion fluxes estimated from absolute measurements of glucose, acetate, and amino acids in the supernatant ($\tilde{v}(t)$) and, only for exchange fluxes with relative estimates of consumption/secretion rates (**u**($t$)), the vector of time-independent scaling factors **c**.

To infer the unknowns **v**($t$) and **c**, we sequentially minimized three objectives: (i) the L1-norm distance between predicted (**v**($t$)) and estimated ($\tilde{v}(t)$) exchange fluxes from absolute metabolite

measurements, (ii) the L1-norm distance between predicted **v**($t$) and estimated relative exchange rates (**u**($t$)), and (iii) the sum of absolute fluxes along the entire time course ($\sum_t |v(t)|$). Assembling the set of constraints generated at each time point in one global optimization problem allows us to solve the vector of scaling factors (**c**) and time-dependent fluxes (**v**($t$)) at once, such that the flux solutions for each time interval depend on each other.

To test uncertainty in flux estimates we performed flux variability analysis (FVA) of the optimal flux solution: for each reaction, we calculated the maximum and minimum sum of fluxes over time. We found that for the vast majority (81%) of metabolic reactions, flux variability was within 10% of the reported flux solution (Supplementary Fig. 6). 738 reactions (29%), out of the 2583 in the model, were active throughout most of the time course, mainly representing reactions essential for energy and biomass generation. Of particular interest are those reactions (8%) that were only transiently active and hence may reflect specific adaptive mechanisms underlying changes in nutrient availability (Fig. 2a). Most of these reactions were involved in transport, exchange and catabolism of amino acids, nucleotide precursors, and intermediates of central metabolism (Fig. 2b).

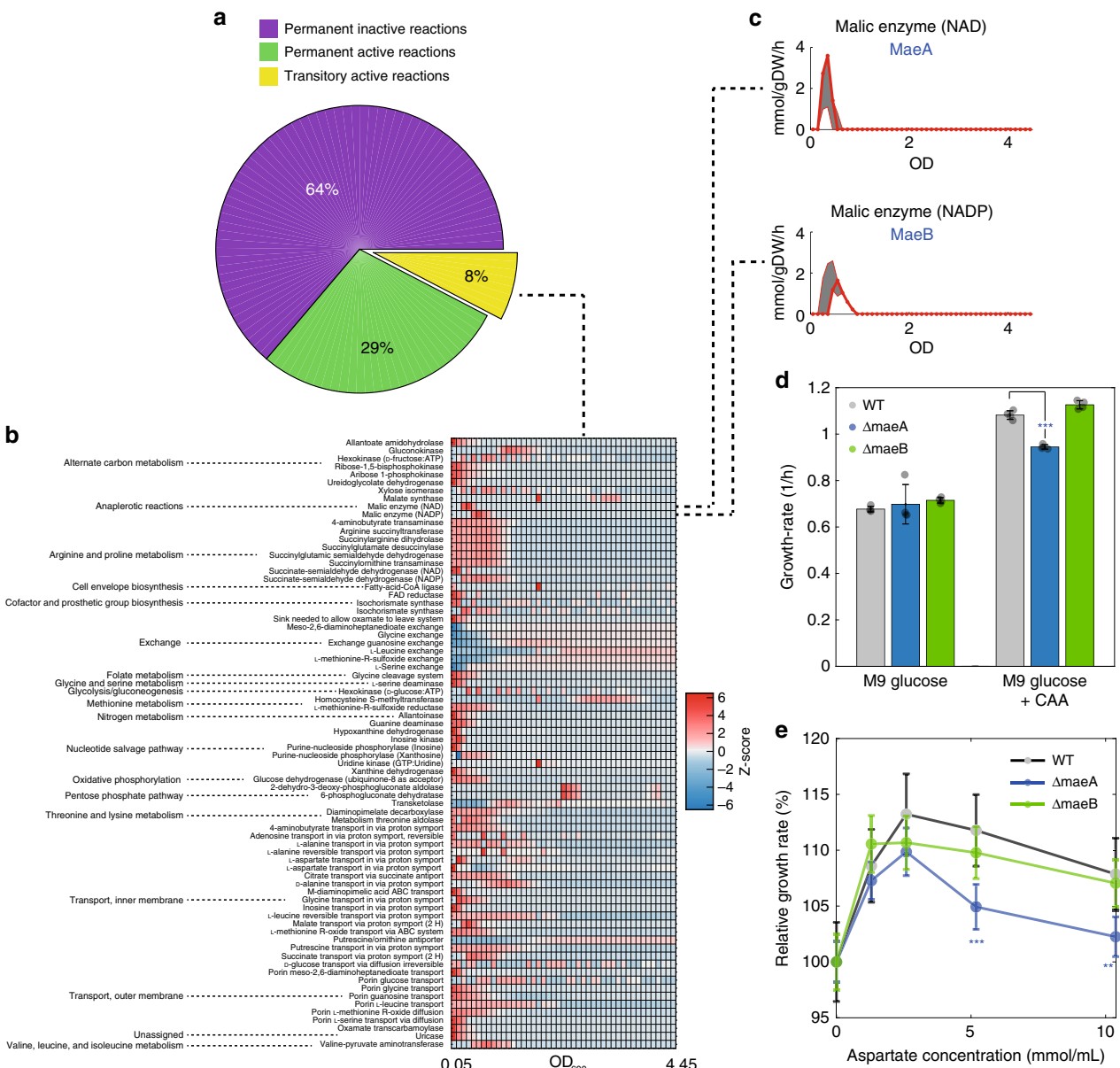

**Fig. 2** Time-dependent prediction of metabolic fluxes. **a** Percentage of inactive (purple) and active (green) reactions throughout the entire time course and reactions that carry flux for a limited time (yellow). **b** z-score normalized values of time-dependent fluxes for transitory reactions with a predicted flux larger than 1 mmol/gDW/h in at least one time point. **c** Predicted time-dependent fluxes through the NAD- and NADP-dependent malic enzymes MaeA and MaeB, respectively. **d** Mean ± standard deviation across four replicates of growth rates in wild-type *E. coli* (WT), ΔmaeA and ΔmaeB in M9 glucose minimal medium and M9 + CAA (***p-value ≤ 0.001, two-tailed paired t-test). **e** Mean ± standard deviation across four replicates of growth rates in wild-type *E. coli* (WT), ΔmaeA and ΔmaeB in M9 glucose minimal medium with increasing aspartate concentrations relative to M9 without aspartate (**p-value ≤ 0.01, two-tailed paired t-test)

Among the transiently active reactions we found the NAD- and NADP-dependent malic enzymes, which catalyze the anaplerotic reaction converting malate into pyruvate (Fig. 2c and Supplementary Fig. 7). While these anaplerotic reactions are inactive in glucose minimal medium[34], time-dependent estimates of flux through the malic enzymes MaeA and MaeB suggest an early transient activation of the flux from malate to pyruvate in the presence of CAA (Fig. 2c). To validate this prediction and test the functionality of malic enzymes in complex medium, we monitored growth of the two individual malic enzyme knockout mutants ΔmaeA and ΔmaeB, in M9 + CAA. Consistent with model predictions of an earlier and stronger activation of the NAD-dependent MaeA enzyme (Fig. 2c), we observed that the maximum growth rates were similar to wild-type in glucose M9,

while ΔmaeA exhibits a significant lower maximum growth rate in M9 + CAA (two-tailed paired t-test, p-value ≤ 0.001) (Fig. 2d). Since malate is close to the entry point of aspartate/asparagine and aspartate is a known allosteric activator of both malic enzymes[35,36], we hypothesized that malic enzymes are particularly important in mediating the utilization of C4-substrates like aspartate. To test this possibility, we repeated the growth assay in M9 glucose medium supplemented with different aspartate concentrations (Fig. 2e). This experiment confirmed our hypothesis and showed that, in the presence of aspartate, ΔmaeA has a reduced growth rate with respect to the wild-type. Overall, we demonstrated that constraining an FBA model with a combination of absolute and relative measurements of metabolite concentrations in the supernatant allows to estimate dynamic

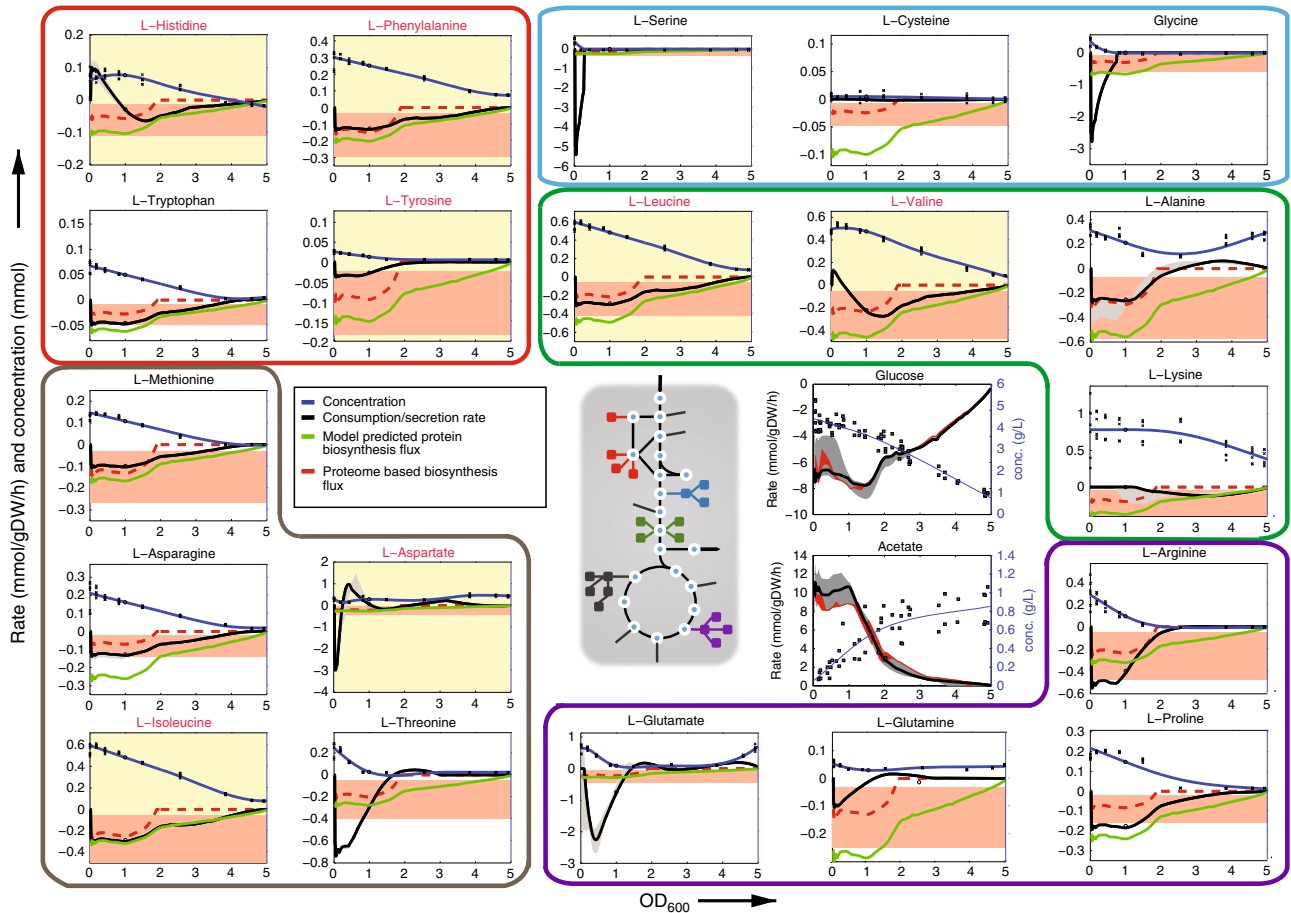

**Fig. 3** Estimated rates of amino acid uptake. Measured amino acid concentrations in the supernatant (crosses) and interpolated values (blue lines). The black line represents the calculated instantaneous uptake rate, and the gray shaded region the confidence intervals. The flux requirements for protein biosynthesis (green line) was estimated from the stoichiometry of biomass composition multiplied by the instantaneous growth rate. Red shaded regions delineate lower and upper bounds of amino acid requirements for protein biosynthesis estimated from experimental measurements of protein abundance in *E. coli* grown in rich medium (~ 20 min doubling time) and glucose minimal medium (~ 60 min doubling time)[43]. Dashed red lines are direct linear interpolations from the upper and lower bounds multiplied by the instantaneous growth rate. The experimentally determined glucose uptake and acetate secretion rates (black line/gray region) are compared to estimates from variability analysis of constrained-based model simulations (red region). The chart backgrounds of all non-degradable amino acids are highlighted in yellow and the font color is red. Amino acids degraded into the same product are grouped by different colors, according to the central schematic metabolic network

intracellular flux rearrangements during the sequential utilization and depletion of nutrients in a complex medium.

**Dynamic coordination of amino acid and glucose consumption**. Our model-based analysis revealed mainly two phases of growth, approximately before and after the culture reaches an $OD_{600}$ of 1. The first phase is characterized by catabolism of amino acids such as aspartate, glycine, glutamate and serine, that provided a large portion of carbon (~40%) and nitrogen (70-80%) for rapid growth during the first ~3 h (Fig. 1d and Supplementary Figs. 8 and 9). During this phase, the culture exhibited low-glucose uptake, high acetate overflow, and an excess consumption of nitrogen (Figs. 1d and 3 and Supplementary Figs. 8 and 9). This excess consumption of nitrogen is predicted to be balanced by secretion of ammonia, a phenomenon also observed in the presence of large quantities of glutamine[37]. Nearly 20% of the carbon required for biomass formation was derived from metabolites other than amino acids and glucose, but no individual metabolite contributed more than 2% to the total carbon balance (Fig. 1d). Notably, in the very first growth phase, model-based analysis of exo-metabolome data predicted high catabolic fluxes

between 2 and 5 mmol/gDW/h for glycine and serine into pyruvate and for aspartate into oxaloacetate (Supplementary Fig. 10). Upon near depletion of these amino acids, glucose and ammonia became the main carbon and nitrogen sources, respectively (Fig. 1d and Supplementary Fig. 9). In this second phase, glucose uptake increased from about 6 to 8 mmol/gDW/h and acetate secretion decreased by approximately 35% (Fig. 3).

Throughout these two phases, the rates of amino acid consumption varied approximately one order of magnitude (Fig. 3). Uptake rates for all seven non-degradable amino acids (Fig. 3, yellow chart areas) were lower than or matched the theoretical requirements for protein biosynthesis at a given growth rate. Akin to uptake of methionine that is inhibited by internal methionine levels[38], it appears that uptake rates of all non-degradable amino acids are tightly regulated by internal feedback loops, possibly to avoid accumulation of toxic intermediates[39]. In contrast, more than half of the degradable amino acids (i.e., 7 out of 13), in particular serine, glycine, threonine, aspartate, and glutamate, were consumed at rates much higher than required solely for protein biosynthesis. Consistent with previous evidence[40], we found that the average amount of amino acid consumed per unit change in $OD_{600}$ (i.e.,

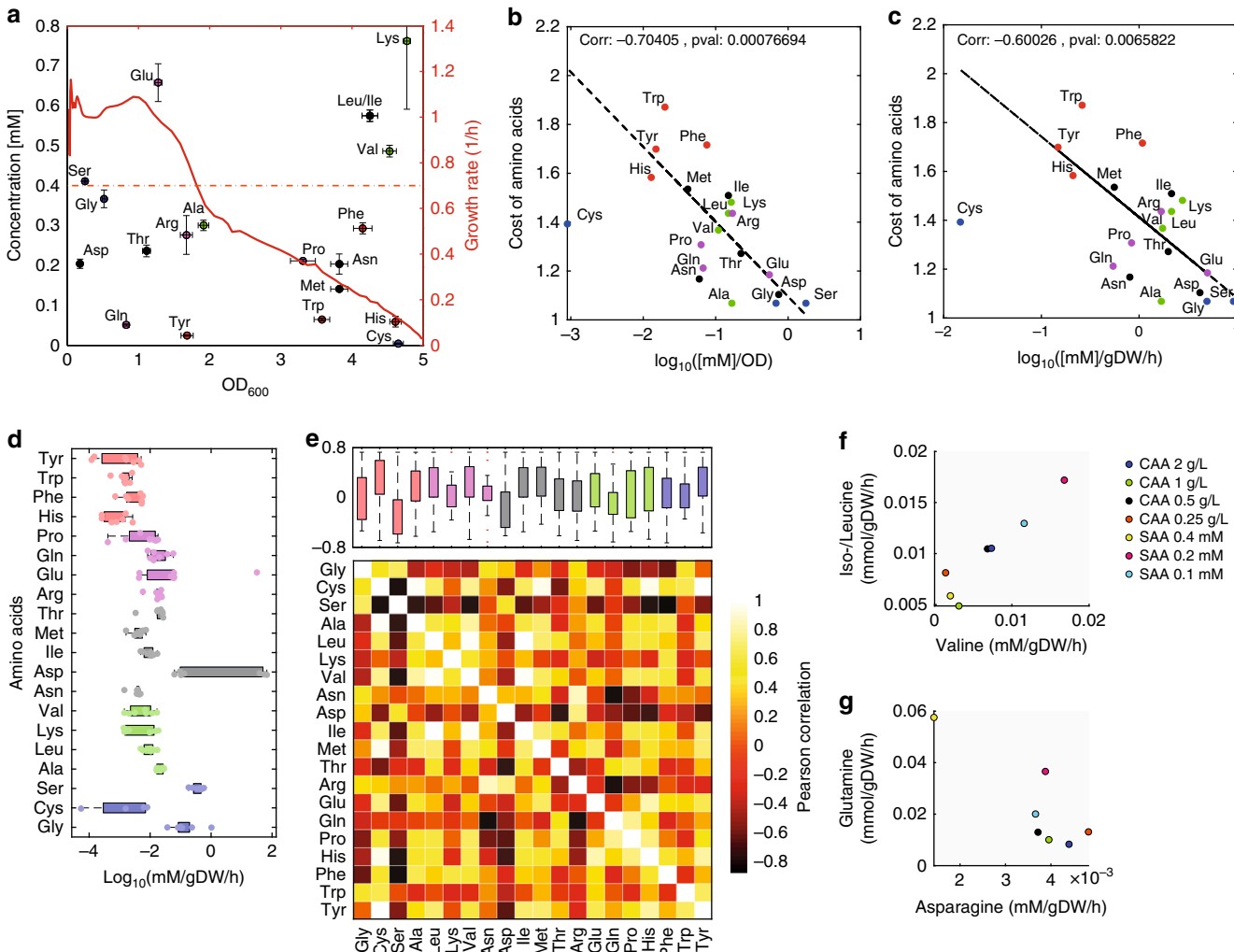

**Fig. 4** Consumption *vs.* cost of amino acids. **a** Each dot represents one amino acid, while the red line is the instantaneous cellular growth rate. For each amino acid, the initial concentration is related to the $OD_{600}$ at which the amino acid has been depleted from the medium. **b** For each amino acid, its metabolic cost, i.e., the number of high-energy phosphate bonds required for biosynthesis[41], is compared to the ratio between initial amino acid concentration and culture $OD_{600}$ at time of depletion shown in **a**. **c** For each amino acid, its metabolic cost[41] is compared to an average estimate of its uptake rate, calculated as the initial amino acid concentration divided by the hours and culture gram of dry biomass (gDW) at time of amino acid depletion. **d** Distribution of amino acid average uptake rates across seven media containing different initial quantities of amino acids (see also Supplementary Fig. 12). **e** Heatmap of pairwise correlation between average uptake rates of amino acids across seven tested conditions. Boxplot of pairwise correlation for each amino acid. Box edges correspond to 25th and 75th percentiles, whiskers include extreme data points, and outliers are shown as red plus signs. **f** Average uptake rates of valine against iso-/leucine. **g** Average uptake rates of asparagine against glutamine

initial amino acid concentration divided by the $OD_{600}$ at time of depletion) is inversely related to its metabolic cost, defined as the number of high-energy phosphate bonds required for biosynthesis[41] (Fig. 4a, b). Our data hence supports the hypothesis that amino acid biosynthetic cost imposes a selective pressure to encode less costly amino acids in highly abundant proteins[41,42] (Supplementary Fig. 11). To test whether the observed amino acid uptake rates depend on their absolute or relative concentration in the medium, we supplemented the medium with seven mixes of all amino acids at different quantities and determined their individual average uptake rates (Fig. 4d). For most amino acids, varying amounts of their concentrations in the media did not affect our previous conclusions (Supplementary Figs. 11 and 12). However, the average uptake rates of some amino acids (e.g., glutamate, glycine, and aspartate) exhibited higher variance than others across conditions (Fig. 4d and Supplementary Data 2). We sought to find potential regulatory dependencies between amino acids by correlating uptake rates between all pairs of amino acids

across the seven conditions (Fig. 4e–g). On average, serine and aspartate exhibited the strongest correlation with uptake rates of other amino acids, suggesting for a prominent role of these amino acids in the regulation of nutrient consumption (Fig. 4e). We found the strongest positive correlation between valine and iso-/leucine, and confirmed the functional relationship between these amino acids by showing that if leucine or isoleucine are depleted before valine, bacterial growth is strongly inhibited (Fig. 4f and Supplementary Fig. 13), in agreement with previous findings[39].

Overall, our data suggests that amino acids that are "cheap" to produce, such as serine, glycine, threonine, and aspartate, are taken up at a much higher rate and degraded, thereby potentially reducing the need for glucose as a carbon and energy source (Fig. 1d). Noteworthy, several low-cost amino acids that were consumed at a rate not exceeding the requirement for protein biosynthesis, namely proline, arginine, glutamine, and asparagine, cannot be directly degraded, but first need to be converted into glutamate or aspartate. The remaining amino acids that could be

directly degraded but are instead taken up at a relatively low rate, are alanine, tryptophan, and lysine. However, differently from the other amino acids, alanine is also an essential component of cell wall peptidoglycan, and tryptophan and lysine have high biosynthetic costs (Fig. 4b, c). Altogether these results suggest for a complex tradeoff between the cost of degrading amino acids—i.e., the risk of wasting expensive resources such as methionine[43] or tryptophan[41]—and the potential benefit to fulfill the demand for carbon and nitrogen.

**The role of pyruvate in coordinating glucose catabolism.** Above, we found that some amino acids are catabolized and even reduce glucose catabolism during the early stages of growth. While glucose-based catabolite repression of less preferred substrates is relatively well characterized in *E. coli*[40,44–46], much less is known about the influence of other nutrients on glucose consumption[47,48]. Thus, we next investigated how *E. coli* coordinates catabolism of amino acids and glucose. As a key regulator of carbon uptake and catabolism, the transcription factor Crp regulates the expression of many alternative substrate uptake systems and genes involved in amino acid degradation and carbon catabolism in *E. coli*[44]. By measuring Crp activity with a GFP reporter plasmid, we verified that glucose strongly represses Crp activity[44] and that the addition of amino acids does not influence this repression (Fig. 5a). Thus, our results suggest that transcriptional regulation by Crp is not responsible for the reduced glucose consumption.

An alternative mechanism for more rapid control of glucose uptake relies on changing transporter activity either via phosphorylation[49] or small molecule binding[5]. Dephosphorylation of the first step of the sugar–phosphoenolpyruvate phosphotransferase system (PTS), EIIA$^{Glc}$, leads to the transport inhibition of several non-PTS carbon sources. According to the current model, a rapid increase of the ratio between phosphoenolpyruvate and pyruvate levels would correspond to increased phosphorylation of EIIA$^{Glc49}$, and hence a reduced glucose uptake. We monitored immediate changes in the intracellular ratio between phosphoenolpyruvate and pyruvate after supplementation of CAA, using a targeted Liquid Chromatography–Mass Spectrometry (LC–MS) method[50] (Fig. 5b, Supplementary Fig. 14, and Supplementary Data 2). We observed a rapid increase of intracellular pyruvate and almost steady levels of phosphoenolpyruvate (Fig. 5b), which according to the current working hypothesis would cause EIIA$^{Glc}$ dephosphorylation and favor glucose uptake[49]. While we do not have direct experimental evidence, reduced phosphorylation of EIIA$^{Glc}$ seems implausible because EIIA$^{Glc}$ is already completely dephosphorylated in the presence of glucose[49]. Moreover, even higher EIIA$^{Glc}$ dephosphorylation would correspond to an additionally increased glucose uptake, contrary to our observation. Thus, collective evidence suggests that the coordination between glucose and amino acid catabolism is achieved by intracellular signaling metabolites; the most parsimonious explanation being modulation of glucose uptake through degradation products of amino acid catabolism.

To test this hypothesis and to identify putative effector metabolites, we supplemented *E. coli* cultures during mid-exponential growth on glucose minimal medium with eight different amino acid mixtures. Each mixture was deprived of one class of amino acids that are catabolized into either of the three final degradation products, namely the α-keto acids: pyruvate, α-ketoglutarate, or oxaloacetate. In each experiment, we determined the rates of growth, glucose uptake, and acetate secretion and measured the dynamic intracellular metabolome response up to

90 min after amino acid supplementation by non-targeted FIA-TOFMS[51] (Supplementary Fig. 15 and Supplementary Data 2). Generally, addition of amino acids affected glucose consumption and acetate secretion (Table 1) and caused large concentration changes in intermediates of central metabolism, primarily pyruvate, and oxaloacetate (see Supplementary Discussion and Supplementary Fig. 15). Relative concentration changes determined by FIA-TOFMS were consistent with previous absolute concentration measurements by LC–MS (Supplementary Figs. 14–16). To identify candidate metabolites that can potentially regulate glucose catabolism, we correlated relative metabolite changes 15, 30, 60, and 90 min after amino acid supplementation with growth rate, glucose uptake rate, acetate secretion rate, and the split ratio of acetate secretion vs. glucose consumption. For at least one time point, four metabolites correlated with the fraction of secreted acetate relative to consumed glucose, eight with acetate secretion, and one with growth rate, respectively (Supplementary Fig 17 and Supplementary Data 2) (absolute fold change ≥ 2 and correlation *p*-value ≤ 0.001). While in CAA supplemented cultures the concentration of most of these metabolites increased steadily over the course of 90 min (Supplementary Fig. 15), pyruvate levels peaked after 30 min and exhibited the strongest correlation with the ratio between acetate secretion and glucose uptake across all supplemented amino acids mixtures (Fig. 5c, d). We found a similar correlation in previously published data[52] monitoring metabolite and flux changes in gene deletion mutants (Supplementary Fig. 18), suggesting that the coupling between pyruvate levels and carbon uptake/metabolism generalizes beyond the specific conditions tested here.

The rapid accumulation and subsequent depletion of intracellular pyruvate levels upon CAA supplementation were compatible with the initial growth phase during which glucose consumption was reduced, and the subsequent phase, after depletion of low-cost amino acids, characterized by increased glucose uptake and decreased acetate secretion (Figs. 4a and 5c). In media lacking amino acids such as serine, glycine, threonine, tryptophan, cysteine, and alanine that are degraded into pyruvate, we observed no suppression of glucose uptake (Table 1), showing that pyruvate levels change in response to catabolism of amino acids. Moreover, previous results from chemostat experiments have shown that increasing glucose uptake corresponds to increased intracellular levels of pyruvate[53], which is opposite to the negative correlation (i.e., Pearson correlation = −0.7) found here between pyruvate levels and glucose uptake. Hence, pyruvate changes are unlikely to be a mere indirect consequence of changes in glucose uptake.

Collected evidence suggests pyruvate as a candidate for the regulation of glucose uptake and acetate secretion. Activation of acetate secretion by pyruvate was already known because pyruvate is a strong activator of phosphotransacetylase, catalyzing the reversible interconversion of acetyl-CoA and acetyl phosphate[54]. Furthermore, *E. coli* grows relatively slow on pyruvate, but with very high acetate secretion of >30% of the consumed carbon[34] and similar overflow metabolism has been observed in other bacteria[55]. Moreover, pyruvate represses the activity of PdhR, a transcriptional regulator that negatively regulates formation of pyruvate dehydrogenase complex (PDHc). We found that while deletion of *pdhR* doesn't affect growth in M9 + CAA, Δ*pdhR* growth rate is mildly (9%) but significantly reduced in glucose M9 (two-tailed paired *t*-test, *p*-value = 0.0045, Supplementary Fig. 19). These results suggest that PdhR residual activity in glucose M9 is completely abolished when amino acids are supplemented to the medium and intracellular pyruvate levels are increased up to 20-fold. The combined increase of pyruvate levels and decreased PdhR activity can

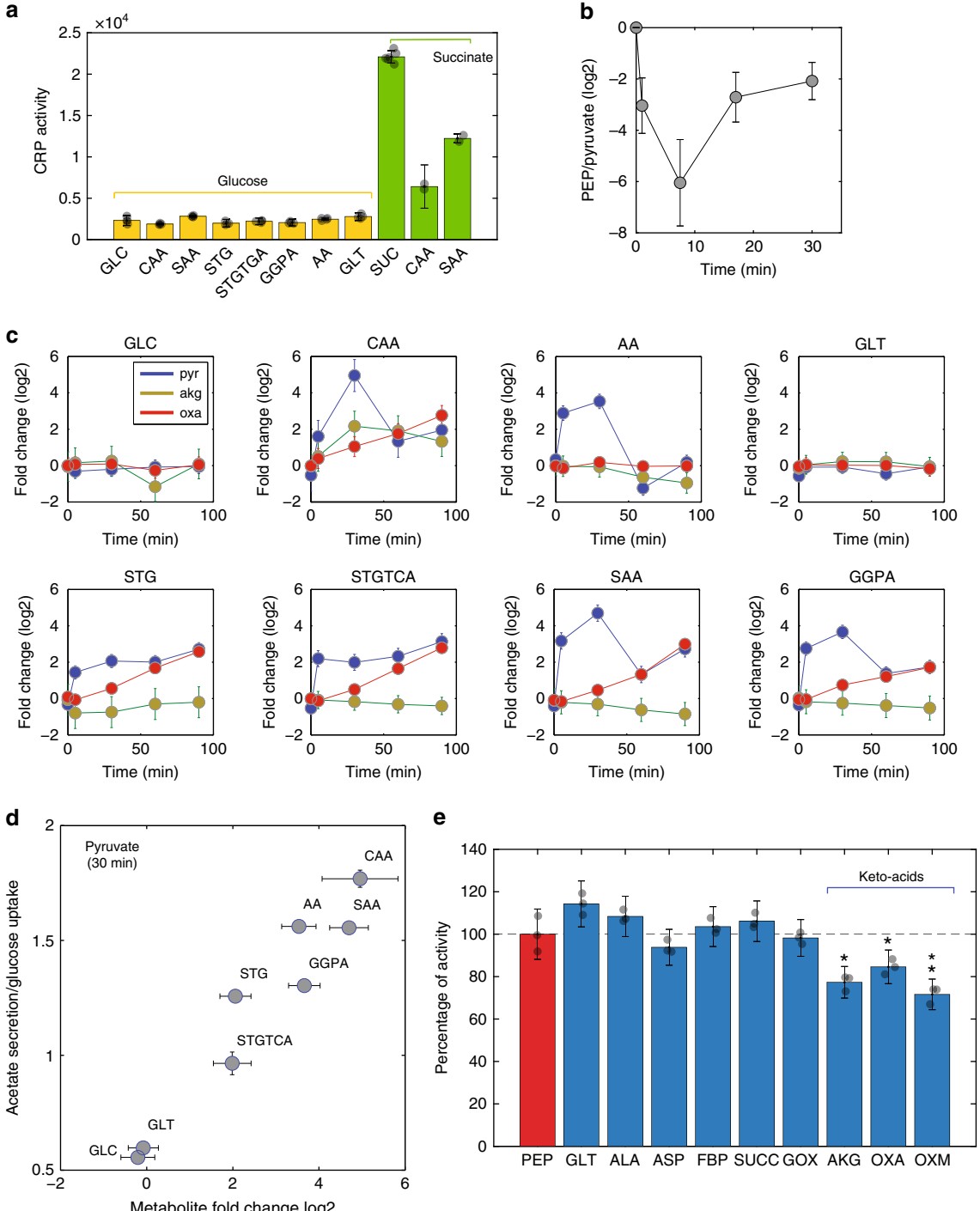

**Fig. 5** Coordination of glucose and amino acids catabolism. **a** Mean ± standard deviation of Crp activity across different nutritional environments: minimal medium with either glucose (GLC) or succinate (SUC in green, positive control), and glucose minimal media supplemented with casamino acids (CAA), synthetic amino acid mix (SAA), SAA deprived of amino acids that can be degraded into pyruvate: (i) threonine, glycine and serine (STG) or (ii) threonine, glycine, serine, tryptophan, cysteine, and alanine (STGTCA), or α-ketoglutarate: (iii) glutamate, glutamine, proline, and arginine (GGPA), or oxaloacetate: (iv) aspartate and asparagine (AA), and glucose minimal medium with 0.125 g/L of glutamate (GLT). **b** Dynamic changes of the ratio between phosphoenolpyruvate and pyruvate upon CAA supplementation. **c** Dynamic relative changes in the abundance of pyruvate (blue), oxaloacetate (red), and α-ketoglutarate (yellow) upon supplementation with the different amino acid mixtures and glucose as the main carbon source. **d** Relative changes in the abundance of pyruvate 30 min after amino acid addition vs. the ratio between acetate secretion and glucose consumption. **e** In vitro activity of PtsI in the presence of only the reactant phosphoenolpyruvate (PEP), or PEP with the addition of glutamate (GLT), alanine (ALA), aspartate (ASP), fructose bisphosphate (FBP), succinate (SUC), glyoxylate (GOX), α-ketoglutarate (AKG), oxaloacetate (OXA), and oxamate (OXM). Bar plots report mean ± standard deviation across three replicates

**Table 1 Growth rate, glucose consumption, and acetate secretion rates of cell cultures grown with M9 glucose medium and SAA mixtures deprived of certain groups of amino acids**

| Condition | Growth-rate (1/h) | GR SD | Acetate secretion (mmol/gDW/h) | AS SD | Glucose uptake (mmol/gDW/h) | GU SD | Acetate secretion/glucose uptake | AS/GU SD |
|---|---|---|---|---|---|---|---|---|
| GLC | 0.70 | 0.03 | −5.43 | 0.16 | 9.13 | 0.61 | 0.59 | 0.043083 |
| CAA | 1.05 | 0.03 | −11.43 | 1.30 | 6.46 | 0.58 | 1.77 | 0.256478 |
| SAA | 1.05 | 0.02 | −11.26 | 0.17 | 7.24 | 0.75 | 1.56 | 0.163309 |
| SAA-STG | 0.80 | 0.02 | −8.92 | 0.36 | 7.10 | 0.65 | 1.26 | 0.126064 |
| SAA-STGTCA | 0.82 | 0.01 | −8.47 | 1.60 | 8.57 | 0.97 | 0.99 | 0.218013 |
| SAA-GGPA | 0.91 | 0.05 | −8.75 | 0.29 | 7.15 | 0.80 | 1.22 | 0.142366 |
| SAA-AA | 1.09 | 0.03 | −8.45 | 0.82 | 5.77 | 0.60 | 1.47 | 0.209082 |
| GLT | 0.79 | 0.03 | −4.60 | 0.28 | 7.71 | 0.32 | 0.60 | 0.044204 |

potentially support a higher flux into acetyl-CoA and possibly acetate.

To test whether pyruvate can also directly regulate glucose uptake, we purified PtsI, the component of the PTS system that catalyzes the phosphorylation of glucose to glucose-6-phosphate with phosphoenolpyruvate (PEP) as phosphate donor. We determined PtsI in vitro activity in the presence of nine different intermediates of central metabolism, using the known inhibitor α-ketoglutarate as a positive control[5,56]. Since pyruvate is one of the reaction products and a readout of the assay, the pyruvate-analog oxamate had to be used to investigate inhibition by pyruvate (Fig. 5e). Consistent with our hypothesis, we found that compounds with similar chemical properties to pyruvate, such as oxamate, oxaloacetate and α-ketoglutarate, but not glyoxylate, inhibited PtsI (two-tailed paired $t$-test, $p$-value ≤ 0.05), while other intermediates of central metabolism such as glutamate, alanine, aspartate, fructose bisphosphate and succinate did not exhibit any inhibitory activity (Fig. 5e). Besides pyruvate's immediate effect on PtsI, analyzing previously published proteomics data[57], we found that pyruvate can induce multiple conformational changes in all protein subunits of the PTS system (i.e., PtsG, Crr, PtsH and PtsI), hinting at the possibility that pyruvate interferes not only with PtsI, but also with the stability of the entire protein complex (Supplementary Fig. 20).

To further demonstrate that increased pyruvate can selectively inhibit the PTS uptake system, we tested for a potential growth inhibitory effect by addition of extracellular pyruvate to E. coli growing with either the PTS carbon source glucose or the non-PTS carbon source succinate. Since Crp-mediated catabolic repression would prevent pyruvate uptake during exponential growth on glucose, we first grew E. coli in glucose M9 media overnight to stationary phase when Crp activity is high, enabling uptake of alternative carbon sources. Overnight cultures from M9 glucose or M9 succinate media were then diluted 1 to 100 in the same minimal media supplemented with 20 or 40 mM pyruvate. Consistent with the pyruvate inhibition of the PTS system hypothesis, we observed that pyruvate addition caused lower growth rates during adaptation to the reappearance of glucose (Fig. 6a). On the contrary, extracellular pyruvate facilitated the adaptation to succinate, resulting in faster growth (Fig. 6b). Consistent with these findings, while succinate consumption is also strongly reduced upon addition of amino acids (Fig. 6c), pyruvate levels exhibit reduced levels (Fig. 6d). Differently from glucose, where Crp activity is already repressed, succinate uptake is directly controlled by Crp[58], and addition of amino acids coincide with proportional reduction in Crp activity (Fig. 5a). Thus, collective evidence suggests that rapid catabolism of serine, glycine and aspartate reduces glucose catabolism and increases acetate secretion via accumulation of α-keto acids, mainly pyruvate and oxaloacetate. On the other hand, while amino acid catabolism modulates glucose uptake via post-translational regulation, transcriptional adaptation seems to be at the basis of global carbon flow regulation in the presence of a non-PTS carbon source.

## Discussion

Resolving dynamic flux variations in continuously changing and nutritionally complex environments has remained a major challenge[59]. Growing in rich media, bacteria cannot consume all nutrients at once, but must decide which to consume first and how to utilize them. Changing nutrient concentrations force bacteria to continuously adapt their uptake, which in turn requires continuous and rapid redirection of intracellular fluxes. Here, we developed an experimental and computational approach that allows to infer dynamic metabolic adaptation phases of E. coli during growth in complex medium. Exo-metabolome patterns revealed complex dynamics in the consumption of nutrients and secretion of metabolites. Somewhat unexpectedly, maximum growth occurred at relatively low-glucose uptake and high acetate secretion. Constraint-based model predictions resolved the overall contribution of consumed nutrients to biomass formation and the intracellular fluxes, unveiling underlying metabolic strategies during the different growth phases with consecutive depletion of amino acids. Low-cost amino acids provided most of the carbon and nitrogen needed during the initial phase of rapid exponential growth that featured relatively low rates of glucose uptake. In particular, serine, aspartate, and glycine were consumed at much higher rates than required for protein synthesis and therefore contributed substantially to energy and biomass generation.

To understand how E. coli coordinates the consumption of different nutrients and adapts its intracellular fluxes, we monitored the intracellular metabolome response to sudden addition of different combinations of amino acids. Pronounced changes in pyruvate levels and their strong correlation with the ratio between glucose consumption and acetate secretion, suggested pyruvate as an important regulator of glucose uptake and acetate secretion. We demonstrated that the pyruvate-analog oxamate inhibits the activity of the glucose uptake system protein PtsI, and that extracellular pyruvate hampers growth resumption of E. coli on glucose, but not on a non-PTS carbon source. Altogether with the previous experimental evidence of pyruvate as an activator of acetate overflow in E. coli[60], our results suggest that catabolism of amino acids and glucose is coordinated through changes in the levels of α-keto acids, primarily pyruvate, and oxaloacetate. Since our experiments were not performed under nitrogen limitation, the minor changes in α-ketoglutarate levels do not contradict previous results on the role of α-ketoglutarate in regulating glucose uptake in response to nitrogen limitation, and extended our understanding on the functional regulatory role of other keto acids in central metabolism, like pyruvate and oxaloacetate.

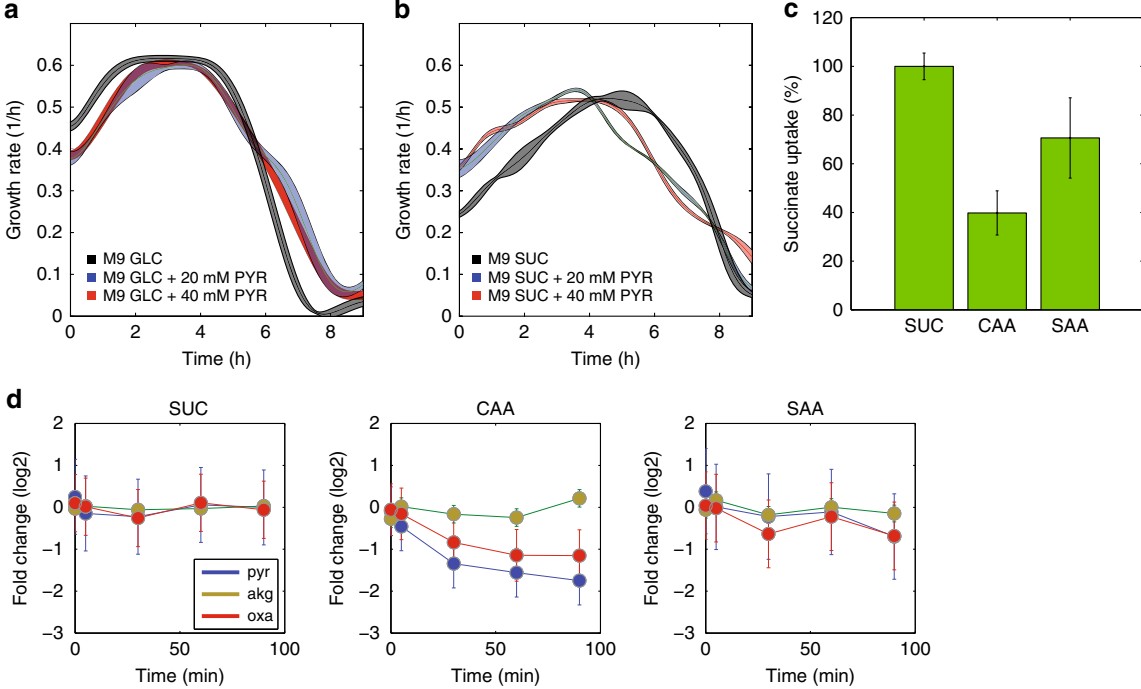

**Fig. 6** Coordination of succinate and amino acid catabolism. **a** Changes in the instantaneous growth rate of *E. coli* in glucose M9 (black), with 20 (blue), or 40 (red) mM of pyruvate. **b** Changes in the instantaneous growth rate of *E. coli* in succinate M9 (black), with 20 (blue) or 40 (red) mM of pyruvate. **c** Relative succinate uptake rates of *E. coli* in succinate M9, supplemented with 2 g/L CAA or a synthetic mix of amino acids (SAA) consisting of the same amino acids and concentrations measured in 2 g/L CAA. Error bars represent the 95% confidence interval from fitting analysis of three biological replicates (see Supplementary Fig. 21). **d** Dynamic relative changes in the abundance of pyruvate (blue), oxaloacetate (red) and α-ketoglutarate (brown) upon supplementation with the amino acid mixtures and succinate as the main carbon source

The presented experimental and computational method can quantitatively model intracellular flux rearrangements during growth of microbes and potentially higher cells in complex environments beyond the here investigated amino acids. We envisage that this new approach coupled with the high time-resolution and coverage achievable with direct infusion mass-spectrometry[24,51,61], in combination with more quantitative methods such as LC–MS or nuclear magnetic resonance[62,63], has the potential to derive testable predictions on regulatory mechanisms that underlie the investigated metabolic phenotypes, even outside of the restricted laboratory conditions.

## Methods

**Strains and media**. For all growth experiments, *E. coli* BW25113 was initially grown for 5 h in Luria-Bertani (LB) medium, and subsequently overnight in M9 minimal medium. As carbon sources 5 g/L glucose and 2 g/L casamino acids (CAA) (Sigma-Aldrich) were used, if not stated otherwise. The M9 medium contained, per liter of deionized water: 7.5 g of $Na_2HPO_4$ $2H_2O$, 3.0 g $KH_2PO_4$, 1.5 g $(NH_4)_2SO_4$, and 0.5 g NaCl and was adjusted to pH 7 before autoclaving. The following components were filter-sterilized separately and then added (per liter of final medium): 1 mL of 1 M $MgSO_4$, 1 mL of 0.1 M $CaCl_2$, 1 mL 0.1 M $FeCl_3$, and 10 mL of a trace element solution containing (per liter) 180 mg $ZnSO_4$ $7H_2O$, 120 mg $MnSO_4$ $H_2O$, 180 mg $CoCl_2$ $6H_2O$, and 120 mg $CuCl_2$ $2H_2O$. Carbon source solutions were filter-sterilized and added separately to the medium.

**Cultivation**. Growth of *E. coli* in M9 minimal medium with glucose and CAA was monitored in a 48-well plate with a Tecan Infinite M200 plate reader (Tecan) (37 °C, linear shaking), by measuring the absorbance at 600 nm. Cell dry-weight was inferred from a predetermined conversion factor of 0.48 g cells/$OD_{600}$[64]. For exo-metabolome profiling, 50 μL of culture broth were sampled at the time points indicated in Fig. 1a and cells separated from the supernatant by centrifugation (4 °C, 4000 rpm, 10 min).

**Exo-metabolome profiling**. Absolute concentrations of glucose and acetate in culture supernatants were determined with enzymatic kits (Megazyme). In parallel, we used a direct flow-injection mass-spectrometry method (FIA-Q-TOF)[21] to profile relative changes of metabolite concentrations in the supernatant.

Specifically, culture supernatants were injected into an Agilent 6550 Series Quadrupole Time-of-flight mass spectrometer (Agilent, Santa Clara, CA), operated in negative mode. The experiment was performed in triplicates and each sample of the ten time points was injected twice. From the spectrum of detected ions, amino acids and other metabolites were annotated as mono-deprotonated ions with the genome-scale model of *E. coli* by Orth et al.[25]. Absolute amounts of amino acids were estimated by comparing peak intensities measured in the culture supernatant with intensities of a dilution series of pure amino acids in M9 minimal medium (see supplementary materials for further details). The validity of the estimated concentrations was confirmed using an alternative standard method for amino acid quantification (ACCQ-Tag Ultra Derivatization Kit, Waters) (see Supplementary Text and Supplementary Fig. 5).Time-dependent profiles of each individual metabolite detected in the supernatant were interpolated using multivariate adaptive regression splines (MARS)[31]. We used the ARESLab Adaptive Regression Splines toolbox for Matlab/Octave ver. 1.13.0, which can be downloaded here: http://www.cs.rtu.lv/jekabsons/. To estimate variability in the fitting estimates for each metabolite, we repeated the fitting 1000 time using only 90% of the collected data. Error estimates were calculated as the standard deviation over the fittings obtained from the bootstrapping procedure. Next, we estimated relative instantaneous uptake and secretion rates by calculating the difference of metabolite levels between two consecutive time points, divide by the time interval (i.e., incremental ratio) (Supplementary Data 1). It is worth noting that an intrinsic problem of data interpolation is the filtering of high frequencies in the data (i.e., smoothing or rapid changes of metabolite levels). In principle other approaches can be used to derive uptake/secretion rates of metabolites in the supernatant, like those described in ref. [65], which can explicitly take into account a priori knowledge on the time at which rapid metabolic changes are expected.

**Amino acid spiking experiments**. To monitor rapid metabolic changes in response to diverse mixes of amino acids and to facilitate dynamic sampling at high-throughput, reducing the risk of sample processing artifacts, we used a 96-well plate whole-culture broth extraction protocol[51]. Aliquots of cell culture were extracted without cell separation using cold solvent extraction and then directly injected into a time-of-flight mass spectrometer[21]. We supplemented exponentially growing *E.coli* cells with synthetic amino acid (SAA) mixtures, containing the same concentrations of individual amino acids as measured in CAA, but deprived of groups of amino acids catabolized into pyruvate (e.g., threonine, glycine, and serine (SAA-TGS) or threonine, glycine, serine, tryptophan, cysteine and alanine (SAA-TGSTCA)) and other α-keto acids such as α-ketoglutarate (e.g., glutamate, glutamine, proline, and arginine (SAA-GGAP)) and oxaloacetate (e.g., aspartate and

asparagine (SAA-AA)). Pure M9 glucose medium containing 0.125 g/L of glutamate (SAA-GLT) was used as a control.

To validate that whole-cell broth (WCB) measurements were representative for intracellular metabolome changes, we also used a fast filtration protocol[4] to exclusively monitor intracellular changes upon CAA and SAA perturbations and showed that most of the patterns we obtained in WCB samples resembled measured intracellular changes (Supplementary Data 2).

For the spiking experiments, *E. coli* was cultivated in 150 mL M9 minimal medium with glucose in 1 L shake flasks (37 °C, 300 rpm) to exponential phase ($OD_{600}$ of 0.2–0.3) before splitting the culture into equal portions for further cultivation (30 mL per 500 mL shake flask and 1.2 mL per well for 96-deep well cultivations, respectively). At an $OD_{600}$ of 0.5, the cultures were spiked with amino acid mixtures to a final concentration of 2 g/L. Subsequently, cell growth was monitored photometrically and a culture volume equivalent to 2 mL at $OD_{600} = 1$ was sampled for intracellular metabolite measurements using the fast filtration technique[4].

For extraction of intracellular metabolites, the filter with cells was transferred into precooled (−20 °C) extraction solution (acetonitrile/methanol/water; 2:2:1) supplemented with 25 μM phenylhydrazine for derivatization of α-keto acids[50]. Only for targeted analysis, 100 μL of an *E. coli* $^{13}C$ extract were added as internal standard to the extraction solution[66], immediately after filter addition. After incubation at −20 °C for 1 h, samples were immediately dried under vacuum at 30 °C for targeted analysis by LC–MS[62] or directly used without drying for untargeted analysis by FIA-Q-TOF[21,51], respectively. For targeted analysis by LC–MS, dried samples were resuspended in 100 μL of water of which ten μL were injected on a Waters Acquity UPLC with a Waters Acquity T3 end-capped reverse phase column (150 × 2.1 x1.8 mm; Waters Corporation, Milford, MA, USA). Metabolites were detected on a tandem mass spectrometer (Thermo TSQ Quantum Triple Quadropole with Electron-Spray Ionization; Thermo Scientific, Waltham, MA, USA).

**Untargeted constrained flux balance analysis**. Standard FBA couples mass balance, thermodynamic (i.e., reaction reversibility) and capacity constraints to stoichiometric metabolic models. Linear constraints define a space of feasible steady state flux solutions, and the flux distributions that maximize an objective function, typically biomass production (i.e., $v_{growth}$), are selected with the aid of linear programming solvers:

$$\max v_{growth}, \ \min \sum_i^n |v_i|$$
$$Sv = 0 \quad (1)$$
$$L_i \leq v_i \leq U_i$$

where, $v$ represents the vector of fluxes, $S$ the stoichiometric MxR matrix (i.e., M metabolites, R reactions), $Sv = 0$ is the mass balance constraint and $U/L$ are the thermodynamic and capacity constraints. Typically, only one substrate is limiting and maximum growth corresponds to maximum yield. The in-silico medium is defined by allowing the import of a metabolite from the external compartment to the intracellular ones (e.g., cytosolic) by adapting the lower/upper bound of the corresponding reaction. Often, the flux distribution that optimizes biomass production and is the most parsimonious—i.e., minimum sum of absolute values, is selected[67].

The genome-scale model of *E. coli* used in this study accounts for 324 exchange reactions[25]. When the medium becomes complex and the limiting resource is not known, the space of feasible solutions increases and simple objective functions as pure maximization of biomass are no longer appropriate[17,68]. In classical constraint-based modeling (e.g., flux balance analysis), the complexity of the in-silico medium is reflected in undetermined systems, where the sparsity of constraints inevitably cause predictions to be inaccurate and unreliable[69]. To overcome these limitations, uptake rates can be experimentally determined and used as constraints in the model to reduce the space of feasible solutions. Here, we used non-targeted mass-spectrometry, which allows fast, accurate, and sensitive measurements of nearly 60% of metabolites that can be exchanged between the cell and its environment (Fig. 1). However, broad coverage comes at the expenses of only relative and not absolute quantitative readouts. For all 20 amino acids, we obviated to this limitation using external calibration curves derived from pure compounds (full details can be found in the Supplementary). Notably, the two isomers leucine and isoleucine, cannot be distinguished by FIA-TOF analysis. Hence for these two amino acids we estimated only upper bounds. Independent measurements of glucose uptake and acetate secretion were performed using enzymatic kits. To incorporate the direct flow-injection mass-spectrometry measurements of the remaining metabolites in a genome-scale metabolic model of *E. coli*, we implemented an optimization procedure that includes auxiliary unknown variables reflecting the linear proportionality factors between measured MS intensities and concentrations. To this end, we divided the experimentally observed growth time course in N intervals at equidistant ODs, from 0.1 to 4.5 with a 0.1 OD interval step and assumed fluxes within these time intervals to be constant. A quasi-steady state assumption is crucial to reduce problem complexity and maintain linearity in the constraints. For each time interval, the absolute rate of consumption and secretion of glucose, acetate and amino acids are estimated from absolute experimental measurements, together with confidence intervals ($v^{measA}$).

The L1-norm distance between absolute measured fluxes ($\tilde{v}$) and predicted ones ($v$) throughout all time intervals is minimized. Subsequently, we minimize the L1-norm distance between the relative estimates of exchange rates for metabolites without a calibration curve, multiplied by a scaling factor c. To this end we introduce an auxiliary variable D. For metabolites detected to be part of casamino acids a maximum initial amount of 50 μmol was considered, consistently with the overall estimates of most abundant metabolites in casamino acids like GMP, uridine and citrate (see Supplementary Text). Results are robust to small changes of this parameter. For each exchangeable metabolite in the model, the uptake at time t cannot exceed the sum of produced and consumed metabolite up to that time. For those compounds found to be secreted in the medium we derive a lower bound for the corresponding c factor was estimated by assuming that if the metabolite was detected in medium at any point during growth, then the maximum concentration has to be >50 μmol. It is worth noting that all measured rates (absolute and relative) are used as soft constraints, besides growth rate, which is here imposed as a hard constraint:

$$\min \sum_{All} |v|$$
$$(\min) \sum_{t=1}^{T} \sum_{i \in \Psi} |D_i| = obj_2$$
$$(\min) \sum_{t=1}^{T} \sum_{i \in \Omega} |v_i(t) - \tilde{v}_i(t)| = obj_1$$
$$Sv(t) = 0 \text{ for } t = 1 : T$$
$$L_i \leq v_i(t) \leq U_i \text{ for } t = 1 : T$$
$$v_i^{measA}(t) - v_i^{errA}(t) \leq \tilde{v}_i(t) \leq v_i^{measA}(t) + v_i^{errA}(t) \text{ for } i \in \Omega$$
$$v_i(t) - c_i u_i^{measR}(t) - D_i = 0 \text{ for } i \in \Psi \quad (2)$$
$$\sum_{k=1}^{T} \sum_{t=1}^{k} v_i(t) \cdot \gamma \cdot \Delta t \cdot OD(t) \geq -0.05 \text{ for } i \in \Psi \text{ and } i \in CAA$$
$$\sum_{k=1}^{T} \sum_{t=1}^{k} v_i(t) \cdot \gamma \cdot \Delta t \cdot OD(t) \geq 0 \text{ for } i \in \Psi \text{ and } i \notin CAA$$
$$L_{ci} \leq c_i \leq U_{ci}$$
$$L_{ci} = 0.05 / \left( \sum_{t=t_0}^{t_{max}} v_i(t) \cdot \gamma \cdot \Delta t \cdot OD(t) \right) \text{ for } i \in \Psi \text{ and } i \notin CAA$$
$$v_{growth}(t) = \tilde{v}_{growth}(t)$$

Here, $\Omega$ is the set of absolute measured fluxes, ψ is the set of measured fluxes with relative estimated rates, c is the unknown scaling factor, $\Delta t$ is the time interval in hours between two contiguous instances, OD(t) is the optical density measured at time t and $\gamma = 0.413$ is the conversion factor used to transform OD readouts in gram of dry biomass (gDW). Solutions were calculated using the CPLEX IBM optimization software. The three minimizations were carried out one after the other in the order described above. Specifically, after solving the first optimization, the minimum sum of absolute differences between absolute estimated and predicted fluxes ($obj_1$) is used as a constraint in the second optimization. In the second optimization, we minimize the L1-norm distance between relative estimates of metabolite secretion/uptake and predicted fluxes. Finally the objective value ($obj_2$) is used as constraint to find the flux solution that minimizes the sum of absolute fluxes. To find the full range of possible flux values (v(t)) we used flux variability analysis[70]. Specifically, we minimize and maximize the flux through each metabolic reaction under the constraint that solutions has to lie in the 99.9% of the optimal solution (Supplementary Fig. 6). Results from flux variability analysis are reported in Supplementary Data 1 and in Figs. 2c and 3 as gray areas.

**cAMP receptor protein (CRP) activity**. Cultivation of the *E. coli* strain bearing a transcriptional reporter plasmid for monitoring Crp activity, as well as the calculation of promoter activity as the $OD_{600}$ normalized GFP production rate, were performed using a previously described protocol[71,72]. Steady state promoter activities in M9 minimal medium with varying external amino acid composition were determined during the first rapid growth phase window during which the cultures exhibited the maximal growth rate.

**PtsI enzyme assay**. PtsI enzyme assays were performed as previously described by Doucette et al.[5]. PtsI (enzyme I) containing an N-terminal histidine tag was purified from ASKA library strain b2416[73] and its identity verified by SDS page. Purified PtsI was pre-incubated for 5 min at 37 °C with a 1:4 mixture of 2 mM sodium phosphoenolpyruvate (Sigma-Aldrich, ≥ 97%) and sodium pyruvate (Sigma-Aldrich, ≥ 99%), in a buffer containing ~25 mM sodium phosphate adjusted to pH 7 and 2.5 mM $MgCl_2$. The reaction was started by addition of an amount of $^{13}C$-labeled sodium pyruvate (Cambridge Isotope Laboratories, ≥ 98%) equivalent to the amount included in the pre-incubation mixture, yielding a final PEP/pyruvate ratio of 1:8. Approximately 1 μg of purified protein was used in each reaction, and the total reaction volume was 300 μl. Putative inhibitors were included in the pre-incubation mixture at a concentration of 2 mM. At each time point of interest, 20-μl samples were simultaneously quenched and diluted by pipetting into 980 μl of 80% v/v methanol in water at −70 °C. After drying the samples and

resuspension in water, the fraction of $^{13}C$-labeled PEP in the total PEP pool was determined by LC–MS analysis. Similarly to[5], for each time course weighted nonlinear least-squares regression was used to fit the following equation:

$$L(t) = 0.45(1 - e^{kt}) \qquad (3)$$

to the mean and s.e.m. of three experimental replicates. $L$ is the $^{13}C$-labeled fraction of the PEP pool at time $t$, and $k$ is the fitted parameter representing the rate constant of the labeling reaction; 0.45 is the $^{13}C$-labeled fraction of total carbon. The estimated $k$-values of each assay were then compared using $t$-test analysis to identify compounds with a significant inhibitory effect (Fig. 5e).

**Reporting summary**. Further information on research design is available in the Nature Research Reporting Summary linked to this article.

## Data availability
All data discussed in this study can be found in the supplementary tables.

## Code availability
Matlab code to reproduce the FBA flux predictions is available at http://www.imsb.ethz.ch/research/zampieri-group/resources.html.

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

## Acknowledgements

We kindly thank Dimitris Christodoulou for helpful feedback and discussions. This work was supported by a Worldwide Cancer Research (WCR-15–1058) project funding to M.Z., an ETH postdoctoral fellowship to M.Z. and SystemsX.ch project TbX from the Swiss National Science Foundation to U.S.

## Author contributions

M.Z. designed the project. M.H., M.Z., N.F.M., F.H. performed the metabolome experiments. M.Z. analyzed the data, designed, and implemented the constraint-based modeling approach. M.Z., M.H., and U.S. wrote the manuscript.

## Additional information

**Competing interests:** The authors declare no competing interests.

