## [Peer Review File · Nature Communications]

Reviewers' comments:

Reviewer #1 (Remarks to the Author):

Zampieri et al provide a very elegant approach for the analysis of exometabolome data during growth on complex substrates. Essentially they use the exometabolome data and use a modeling framework to estimate (though there is an insinuation of novelty and prediction in the method used) the fluxes at different time points. They then use the fluxes to identify regulation of intracellular reactions dynamically and specifically identify MaeB as a reaction that is upregulated and they show that the deletion of this reaction leads to poorer growth on the complex media used. In addition, they also hypothesize that pyruvate might be a regulator that leads to decreased glucose uptake rate and provide some in vitro support for the idea that pyruvate might regulate PTS. The paper is well written except for the abstract and while the method itself is not arguable novel the results of the metabolic flux measurements and the data certainly are valuable. However, there are a couple of major points that need additional data as outlined below.

Major Concerns:

Line 120:

If the scaling factors are also solved for in addition to the fluxes, then theoretically any concentration can be matched up with a flux using the c_i . This does not seem to be appropriate. The key I guess is that the scaling factors is constant over time. This has to be specified. The optimization specified also does not seem pretty novel. It seems as if the model is flux estimation problem where the fluxes are estimated from the exo metabolome concentrations....In the optimization problem there are three different objective functions, the difference between measured and estimated flux vectors, the scaling factor errors, sum of fluxes. Clearly this is solved using some weights and these details are missing in the formulation. Finally, I assume that the measured fluxes are only glucose and acetate and perhaps the authors can indicate how these measure fluxes are calculated. It is important to clarify what is being measured, what is being calculated from other measurements and what is being estimated using the exometabolome measurements. Regardless, such a formulation is something that flux analysis field is generally familiar with. The flux variability formulation also needs to be incorporated as this can be different from the standard algorithm. It is important to identify fluxes that are subject to noise. For example in figure 2, it is important to know how the activation of reactions is impacted by the noise in the exometabolome measurements. What if a 5 or 10% noise in the metabolite concentration measurements are being introduced through sample prep variations?

Line 243:

What is figure 5D trying to convey? The y axis is labeled as yield where as the caption refers to this as a ratio. The x axis is fold change of pyruvate with the dots being the different conditions. Does this mean that pyruvate concentrations changes more when there is increased acetate uptake as compared to glucose? Even so this is not so intuitive to rationalize and the authors just sweep this along with the rest of the figures.... I think the authors would like to suggest that pyruvate concentrations changes the glucose uptake rate where as they have not discounted the possibility that changes in glucose/acetate could lead to changes in pyruvate. I am not sure whether this data conclusively supports the hypothesis around pyruvate regulation. In addition, the fact pyruvate activates acetate secretion could also explain this graph. Figure 5E is again not totally convincing that pyruvate regulates PTS system. Ideally additionally biochemistry needs to be done to figure out the transport at varying concentrations of pyruvate to tease apart this potential allosteric inhibition to strengthen the paper.

The abstract is not very well written. It speaks of a novel approach but what is not clear is whether this is an algorithm or a framework and this needs to be clarified. The key finding seems to be regulation of PTS by ketoacid Pyr and Akg but is this novel? This needs to be clarified a bit better in the abstract. The title has the word "coordination mechanism" which is pretty awkward and can

be replaced. Finally, there is a mention of maximum instantaneous growth rate that corresponds to reduced glucose uptake and this sentence is out of context and perhaps the authors can add a sentence to explain this better.

Minor Concerns:

Line 38:

There are several nonstationary MFA approaches that have been published and could be cited here.

Line 44:

Some of the references in 17,18,19 do not explicitly mention FBA. There seems to be some biophysical model developed here.... Perhaps correct these refs with others which use FBA ?

Line 56:

Why was casamino acids used ? Given that it is undefined, it would be seem more appropriate to have a more defined substrate for the transitions ?

Line 74:

How is this instantaneous growth rate calculated ? Please refer to a section in either methods or SI and add a sentence clarifying this. This is not clear...

Figure 1B:

Change Intracellular/Extracellular "deteceted" to detected

Line 108:

I am not sure what this phrase means "using an incremental ratio approximation"

Line 114:

There are other references where a similar idea has been looked at. So please cite.
PMID:29222762

Line 208:

The argument about the technical limitations on the measurement of EIIP seems difficult to take.

Figure 3:

Line 672

constrained-based model simulations (red region). Non-degradable amino acids are highlighted in yellow.

Where is the yellow ? May be my eyes are too old but still ???

Figure 4: How come Ser uptake rate is positive ? Is it secreted ?

Line 136:

I find it interesting that Malic enzyme reactions are being predicted. I am wondering whether these are ways in which the cell can help with redox balance associated with NADH and NADPH ? Figure 1B has two shades of gray. It is not clear what those two refer to.

Line 609:

Ref. 57 needs to be reformatted

Fig S4 has a dark blue section that is not defined....

Reviewer #2 (Remarks to the Author):

The present manuscript by Zampieri et al. provides for the first time a description of a platform to investigate the dynamic changes in the intracellular and extracellular metabolome of the model organism *E. coli* facing the typical nutrient limitation in nature. How the shift of the metabolism from "normal" biochemical pathways to the challenging new environmental conditions is taking place is still an open question; even though we know a lot about model organisms like *E. coli*.

With regards to this open questions the present contribution provides new methodology and insights into the dynamic processes taking place during the challenges of nutrient limitations. Not only the analysis of metabolite patterns was undertaken but also selected mutant strain experiments were performed. In particular I like to mention the PTS activity assays using nine different intermediates of the central carbon metabolism to understand the role of pyruvic acid in the adaptation toward environmental stresses. The importance of pyruvate as a candidate metabolite for the regulation of glucose uptake and acetate secretion was well investigated.

An important aspect I see in the clear description of the data without over interpretation and pronounced speculation. From my point of view in particular the assessment of the role of (sic!) "low cost amino acids" in the bacterial growth is a major outcome of the study.

The authors used a non-targeted approach to analyze the metabolome. To further investigate the changes in the amino acid pool they used the ACCQ-TAQ assay. Would it be worth to think about a more comprehensive analysis of the extracellular metabolome using methods like NMR? Of course: the limit of detection of NMR is low and compared to MS limited to the database used for the analysis. Nevertheless the opportunities to elucidate further metabolite patterns should be taken into account.

The method section is compiled in a very consistent way and provides the necessary data to follow the experimental design by the authors.

The manuscript text needs only some minor spelling check; like "filter sterilized" vs. "filter-sterilized" (page 11, 314, 317). I suppose this could be done in the end review.

Overall the contribution will be of great impact to the community and will certainly influence new approaches to understand the adaptation processes in the bacterial cell towards environmental challenges and the close connection of distinct biochemical pathways.

Reviewer #3 (Remarks to the Author):

In this study, the authors characterize *E. coli* metabolism in complex medium, by combining measurements of external metabolites (with non-targeted mass-spectrometry) and a constraint-based modeling approach (a kind of dynamical metabolic flux analysis). The model used is the iJO1366 reconstruction of the *E. coli* metabolism. More precisely, metabolite concentrations are used to determine the secretion and uptake rates along growth. Solving a global optimization problem allows predicting internal metabolic fluxes. Deeper analysis of the predicted fluxes and measured concentrations along growth allows the authors to identify that amino acids are first catabolized, and then only glucose is consumed while acetate is excreted. Ketoacids like Pyruvate are proposed to regulate the coordination between amino acid and glucose catabolism through inhibition of PtsI activity.

Overall, the paper is interesting for the community, clear and well written. However, it is a pity that the text in the artwork (in the main text and the supplementary information) is often so small that it is impossible to fully interpret the figures when printed. Their screen resolution does not permit zooming too much as well. Table S2 is missing, Table 1 is in fact Table S1, and numbering is shifted at some point in the list of references (e.g ref 31 should be 29).

While I found the results convincing in general, I have a number of comments/questions detailed below.

1) Properly estimating external fluxes is crucial in this paper, since most results are derived from them.

a) The authors should provide more details on how they spline fitted the metabolite concentrations and OD600 (beside the small explanation on lines 106-108). They could easily add a section in the supplementary information to describe their approach. Some related concerns follow:

b) The phenomena the authors look at happen in the first few hours of growth, when measurements of optical density and concentrations are especially noisy and difficult to interpolate. Spline fitting is known to be particularly sensitive to noise. Uncertainty in the spline fitting propagates to the uptake/secretion/growth rates due to the spline differentiation (and division by fitted OD in the fluxes). Hence, the curve for the growth rate shown in Figure 1 reflects maybe the reality, but it may also (partially) result from fitting problems.

c) In addition, spline fitting has difficulties to capture sudden changes of concentrations and smooths too much the time-course profiles, which possibly hides transient phenomena. How did the authors deal with this possible problem, given that they are interested in metabolic transitions?

d) Some fits do not follow well the trend of the data. Are there some constraints in the fitting procedure? See for instance the Proline and Alanine profiles: their negative flux indicates some consumption below OD=1, while the data show a plateau instead.

2) Section "Dynamic coordination of amino acid and glucose consumption"

a) lines 181-184: The authors conclude from the amino acid supplementations shown in Figure S11 that the uptake rates do not depend on the amino acid concentration. This is certainly true for Asp, Ser and Gly that are consumed before Glucose, even at 0.25% CAA. However, this is not always true for other amino acids that seem to be co-consumed with Glucose at low CAA concentration, while they are consumed after glucose depletion at higher CAA concentration. See for instance the case of Trp.

b) lines 185-189: Why should the cost of amino acid synthesis drive mostly the order in which amino acids are catabolized? The authors do not fully interpret the supplementary results in Figure S11 (notably the two last columns), but this could be helpful to follow their line of reasoning. What is their interpretation?

c) Same lines. An alternative interpretation to the synthesis cost could be the following one: according to the third column (and if I can read correctly due to the low figure resolution), it seems that amino acids that are consumed first are also among the most needed to generate new cells (see Ser for instance). Therefore, the ordering in the consumption of amino acids and glucose could simply reflect a cell demand. In the same line, and if I understood correctly the y-axis in the last column (how exactly are determined the average relative amino acid concentrations among multiple environments?), these amino acids are a bit less abundant than in many environments, which puts a pressure on cells to import these needed amino acids and to catabolize them in order to grow and divide.

3) Section "The role of pyruvate in coordinating glucose catabolism"

The authors monitored Crp activity but could not involve this factor in the reduced glucose consumption. Did the authors also look for changes in PdhR activity in response to amino acid addition?

4) Section Methods

a) Line 426: why do the authors impose the growth rate as a hard constraint? This could be problematic if there is some uncertainty on the determination of the growth rate (e.g. for the reasons given in comment 1b). Why not trying to minimize the L1-norm distance between the predicted and measured growth rate, along with the uptake and secretion rates?

b) Is the computer code available to allow reproducing the results?

5) Additional comments/questions:

a) The uptake and excretion of metabolites is correlated with the accumulation of biomass. This information could be used in the data spline fitting, instead of fitting the curves separately (see doi: 10.1093/bioinformatics/btx250 for an example). This could constrain the fits more and possibly give more accurate estimations of the rates.

b) What is the need to select the solution minimizing the sum of absolute fluxes in the global optimization problem? The minimization of the distance between measured and predicted external fluxes together with the constraints on fluxes do not suffice to reduce the space of solutions to solutions with physiologically reasonable values?

6) Minor comments and misspellings:

a) Methods: line 428: D_i is not defined

b) Figure 1: the shaded region in panel A is not described in the legend. Misspellings in panel B: detected->detected

c) Figure 5, panel C: akg should be displayed in green in the figure legend.

d) Table 1: add a column with SD on the Acetate/Glucose ratios.

e) Figure S1, right panel: display the SD for the two growth rates.

f) Figure S11: mention that y-axes in the fourth column are on a log scale. Provide more information in legend on how the average relative amino acid concentrations across multiple environments are determined.

g) Same figure, labels A to D are missing.

h) Figure S15, title: glucose->glucose; axis label: Maximu->Maximum

i) Main text, line 177: Figure 3, dashed green line->green line

j) Main text, line 432: in in gram->in gram

k) Main text, bibliography: reference style is heterogeneous.

l) Artwork: improve resolution and text size in Fig 1B, 3, S10, S11

Reviewers' comments:

Reviewer #1 (Remarks to the Author):

Zampieri et al provide a very elegant approach for the analysis of exometabolome data during growth on complex substrates. Essentially they use the exometabolome data and use a modeling framework to estimate (though there is an insinuation of novelty and prediction in the method used) the fluxes at different time points. They then use the fluxes to identify regulation of intracellular reactions dynamically and specifically identify MaeB as a reaction that is upregulated and they show that the deletion of this reaction leads to poorer growth on the complex media used. In addition, they also hypothesize that pyruvate might be a regulator that leads to decreased glucose uptake rate and provide some in vitro support for the idea that pyruvate might regulate PTS. The paper is well written except for the abstract and while the method itself is not arguable novel the results of the metabolic flux measurements and the data certainly are valuable. However, there are a couple of major points that need additional data as outlined below.

We thank the reviewer for the overall positive and encouraging comments. We rephrased the text to avoid any overstatement and to clarify the novel aspects of the computational framework and how model predictions guided the follow ups and main findings of this study.

Major Concerns:

Line 120:

If the scaling factors are also solved for in addition to the fluxes, then theoretically any concentration can be matched up with a flux using the c_i . This does not seem to be appropriate.

They key I guess is that the scaling factors is constant over time. This has to be specified. The optimization specified also does not seem pretty novel. It seems as if the model is flux estimation problem where the fluxes are estimated from the exo metabolome concentrations....

We thank the reviewer for raising this point and giving us a chance to clarify this important aspect. For each exchange reaction, parameter c is a constant value (i.e. independent of time). C is an unknown scaling factor virtually representing the proportionality of signal intensities measured by our non-targeted mass-spectrometry platform and the actual metabolite concentration in the supernatant. Notably, this scaling factor is applied only to the fluxes of exchange reactions in the model (i.e. nutrient uptake or byproduct secretion). Hence, internal fluxes are predicted based on the objective functions of the model (see following point). We clarify this point in the text.

There are two main elements of novelty in our computational approach:

- To our knowledge this is the only framework able to make use of absolute as well as relative measures of uptake and secretion rates, and hence to take advantage of the throughput and coverage of modern high-throughput non-targeted mass spectrometry.
- The above mentioned flux estimation from exo-metabolomics data is the classical dynamic FBA approach that relies only on absolute measurements of few metabolites in the supernatant and typically uses classical objective function (e.g. maximization of biomass yield) to solving the

optimization problem in each time interval independently. Our methodology, in contrast, is formulated as a single global optimization problem. This global optimization is the key to incorporate absolute and relative measures of exometabolome and resolve both time-dependent fluxes and time-independent scaling factors c .

In the optimization problem there are three different objective functions, the difference between measured and estimated flux vectors, D the scaling factor errors, sum of fluxes. Clearly this is solved using some weights and these details are missing in the formulation.

We now clarify this aspect. These objectives are optimized one after the other, so no weights are used. First, we restrict the space of scaling parameters c and flux solutions to those minimizing the distance between predicted and measured uptake rates of glucose and amino acids, secretion of acetate and growth rate. Next this objective becomes a new constraint in the following optimization, where we minimize the distance between uptake-secretion rates of all metabolites measured only in relative amount. Finally, the second objective becomes a new additional constraint in the model. To restrict the space to the most meaningful flux solutions (i.e. avoid solutions with futile cycles), we used a classical minimization of the total sum of absolute fluxes ¹.

Finally, I assume that the measured fluxes are only glucose and acetate and perhaps the authors can indicate how these measure fluxes are calculated. It is important to clarify what is being measured, what is being calculated from other measurements and what is being estimated using the exometabolome measurements. Regardless, such a formulation is something that flux analysis field is generally familiar with.

Point well taken. We now explain the overall experimental and computational procedure in more detail. We emphasize that the metabolites for which we measured absolute concentrations in the supernatant are glucose, acetate and 20 amino acids. For ~400 metabolites we measured relative levels in the supernatant. We also clarify how instantaneous uptake and secretion rates are calculated using the instantaneous derivative in time of interpolated metabolite profiles in the supernatant.

The flux variability

formulation also needs to be incorporated as this can be different from the standard algorithm. It is important to identify fluxes that are subject to noise. For example in figure 2, it is important to know how the activation of reactions is impacted by the noise in the exometabolome measurements. What if a 5 or 10% noise in the metabolite concentration measurements are being introduced through sample prep variations?

We thank the reviewer for this suggestion. We clarify that we use standard FVA analysis and details now can be found in materials and methods. Notably, variability in metabolite measurements depend on multiple factors (i.e. polarity, abundance, ionization properties), and hence can differ among metabolites. We are aware of the importance in considering this aspect, and hence we agree with the reviewer. In our framework, noise in the measurements is carefully taken into account.

Specifically, we formulated the optimization problem to include measurement variability across the biological replicates (reported in Table S1) (see model equations in the materials and methods) so that $\tilde{v}(t)$ are within the exometabolome derived average \pm standard deviation of estimated absolute and relative exchange fluxes. To more specifically address reviewer's comment on Fig. 2 (same apply to Fig. 3) results from flux variability analysis (i.e. how noise in the measurements propagate to flux estimation) are highlighted by the grey dark zones. Hence our final estimates of fluxes already account for noise

measurements (reported in Table S1), and we now more carefully described estimates of measurement variability and how noise in the measurements are considered in the modeling.

Line 243:

What is figure 5D trying to convey ? The y axis is labeled as yield where as the caption refers to this as a ratio. The x axis is fold change of pyruvate with the dots being the different conditions. Does this mean that pyruvate concentrations changes more when there is increased acetate uptake as compared to glucose ? Even so this is not so intuitive to rationalize and the authors just sweep this along with the rest of the figures.... I think the authors would like to suggest that pyruvate concentrations changes the glucose uptake rate where as they have not discounted the possibility that changes in glucose/acetate could lead to changes in pyruvate. I am not sure whether this data conclusively supports the hypothesis around pyruvate regulation. In addition, the fact pyruvate activates acetate secretion could also explain this graph.

Apparently our explanation of this figure was too short. In Fig. 5D we show that 30 minutes after supplementation of different mixes of amino acids, increased pyruvate levels scale with the ratio between acetate secretion and glucose uptake. More specifically, we found that the higher the level of pyruvate, the lower glucose uptake and the higher is acetate secretion. This association becomes significant and rather specific when comparing pyruvate levels with the ratio between acetate and glucose exchange (i.e. only 3 other metabolites exhibit comparable correlation Fig. S15). We do indeed agree with the reviewer that such correlation doesn't reveal causation. While the direct regulation of pyruvate on acetate secretion has been experimentally demonstrated², its direct role in mediating glucose uptake remained to be clarified. We have multiple lines of evidence suggesting a direct role of pyruvate in mediating glucose inhibition and performed additional experiments to reinforce our findings:

- Firstly, pyruvate direct inhibit PTSI in vitro (Figure 5). Notice that intracellular concentrations range from 0.25-0.39 mM (i.e. reported basal level of pyruvate in glucose M9^{3 4}) to ~8mM (in glucose M9 +CAA where we have almost 20 fold increase), and are hence within the range of the 2mM concentration tested in the in vitro assay (please see also point below).
- We observe that if we don't supplement those amino acids degradable in pyruvate, we don't see any pyruvate accumulation and reduction in glucose uptake (Fig. 5).
- To support our conclusions that pyruvate is selectively interfering with the uptake and metabolism of PTS carbon sources, we performed new experiments.
 - o We show that pyruvate inhibits growth of *E. coli* in a glucose M9, while an opposite beneficial effect is observed when *E. coli* is grown on a non-PTS carbon source like succinate (new Fig. 6).
 - o Next, we investigated whether amino acid supplementation also affects the consumption of non-PTS carbon sources, and if yes, what are the underlying mechanisms. To address this question, we grew *E. coli* cultures on succinate as the sole carbon source and repeated the amino acid supplementation experiments. We showed that also succinate uptake is inhibited by amino acids supplementation. However, differently from glucose, pyruvate levels are not affected and coordination between catabolism of succinate and amino acids associates to large changes in Crp activity (new Fig. 5-6).

Figure 5E is again not totally convincing that pyruvate regulations PTS system. Ideally additionally biochemistry needs to be done to figure out the transport at varying concentrations of pyruvate to tease

apart this potential allosteric inhibition to strengthen the paper.

The reviewer has a point. We inappropriately used the word allosteric in the main text. As a matter of fact the reviewer is correct and with the presented data we cannot rule out whether pyruvate acts a competitive or allosteric inhibitor. We correct the text to avoid any overstatement. Indeed NMR experiments^{5 6} nicely clarify the molecular mechanisms (i.e. competitive inhibition) underlying the inhibition of PTSI from α -ketoglutarate described in ⁷. In our case, the fact that pyruvate is also a substrate of the enzyme complicates the biochemical validation, as we expect pyruvate to naturally bind to PTSI.

Hence, we decided to reanalyze previously published limited proteolysis data⁸, systematically mapping conformational changes in the *E. coli* proteome upon incubation with 20 different metabolites, including pyruvate. Not only we found that PTSI undergoes significant ($q\text{value} \leq 0.01$) conformational changes proximal to the PEP binding site (Figure S20), but that all protein subunits of the PTS system (PtsG, Crr, PtsH and PtsI) exhibit conformational changes on multiple sites. Overall these results hint at the possibility that pyruvate plays a key role also in the stability of the complex. (Please also see comment above)

The abstract is not very well written. It speaks of a novel approach but what is not clear is whether this is an algorithm or a framework and this needs to be clarified.

We did not explicate adequately the key novelties of our method and results. The originality of our approach stands from its ability to incorporate dynamic non-targeted MS data in FBA models to derive estimates of intracellular dynamic flux rearrangements upon continuous changes in nutrient composition. Two are the key advantages of this combined experimental/computational approach: (i) using our non-targeted MS platform we can detect a large number of potential nutrients/byproducts and eventually increase time resolution up to seconds⁹, (ii) by reformulating the FBA problem as a single optimization problem and introducing new variables (i.e. parameters c) we can use relative measurement of metabolite abundance in a mass balance framework.

The key finding seems to be regulation of PTS by ketoacid Pyr and Akg but is this novel? The needs to be clarified a bit better in the abstract.

While new technologies advanced our ability to find new regulatory interactions between metabolites and proteins^{8 10}, resolving the functional role of such interactions remains a major challenge. Inhibition of glucose uptake by keto acids, specifically pyruvate, has been indirectly observed already in the 70s^{11 12}. The key novelty in our results is the direct evidence from in vitro enzyme assays that pyruvate can inhibit PTSI activity and, even more important, the functional role of this regulatory interaction in mediating the utilization of glucose and amino acids. We now clarified these aspects in the main text.

The title has the word "coordination mechanism" which is pretty awkward and can be replaced. Finally, there is a mention of maximum instantaneous growth rate that corresponds to reduced glucose uptake and this sentence is out of context and perhaps the authors can add a sentence to explain this better.

We modified the title and extensively revised the abstract to avoid confusion.

Minor Concerns:

Line 38:

There are several nonstationary MFA approaches that have been published and could be cited here.

OK

Line 44:

Some of the references in 17,18,19 do not explicitly mention FBA. There seems to be some biophysical model developed here.... Perhaps correct these refs with others which use FBA ?

OK

Line 56:

Why was casamino acids used ? Given that it is undefined, it would be seem more appropriate to have a more defined substrate for the transitions ?

The choice was intentional because we aim to study more natural environments where “media composition” is undefined and complex. Casamino acids were a perfect choice being both complex and undefined with many non amino acid compounds.

Line 74:

How is this instantaneous growth rate calculated ? Please refer to a section in either methods or SI and add a sentence clarifying this. This is not clear...

Full details are now included in the main text when describing the dynamic FBA approach.

Figure 1B:

Change Intracellular/Extracellular “deteceted” to detected

OK

Line 108:

I am not sure what this phrase means “using an incremental ratio approximation”

We rephrased the text to clarify this point

Line 114:

There are other references where a similar idea has been looked at. So please cite. PMID:29222762

Thanks for the suggestion. We thought the original paper : “DFBALab: a fast and reliable MATLAB code for dynamic flux balance analysis” was more appropriate.

Line 208:

The argument about the technical limitations on the measurement of EIIP seems difficult to take.

We rephrased that sentence

Figure 3:

Line 672

constrained-based model simulations (red region). Non-degradable amino acids are highlighted in yellow.

Where is the yellow ? May be my eyes are too old but still ???

We now highlighted these amino acids using a different color for the labels.

Figure 4: How come Ser uptake rate is positive ? Is it secreted ?

The reviewer is right and positive uptake rates represent amino acid secretion, which we detected for some amino acids while approaching stationary phase. Interestingly we found that many intermediates of low cost amino acids are also secreted when cells approach stationary phase (Fig. S4). On the contrary, metabolic intermediates of expensive amino acids like methionine were not secreted at any time during growth.

Line 136:

I find it interesting that Malic enzyme reactions are being predicted. I am wondering whether these are ways in which the cell can help with redox balance associated with NADH and NADPH ? Figure 1B has two shades of gray. It is not clear what those two refer to.

This is a very good point, indeed we thought about this possibility as well. While malic enzymes have been reported to contribute to redox balancing in *B. subtilis*¹³, we are not aware of data supporting such a role of malic enzymes in *E. coli*, where dedicated transhydrogenases UdhA and PtnAB are present. Hence, because we don't have direct evidence supporting the role of malic enzymes in mediating redox balance we refrain from any speculation.

We now clarify that the shaded region represents the uncertainty in flux predictions from the Flux Variability analysis.

Line 609:

Ref. 57 needs to be reformatted

OK

Fig S4 has a dark blue section that is not defined....

OK

Reviewer #2 (Remarks to the Author):

The present manuscript by Zampieri et al. provides for the first time a description of a platform to investigate the dynamic changes in the intracellular and extracellular metabolome of the model organism *E. coli* facing the typical nutrient limitation in nature. How the shift of the metabolism from "normal" biochemical pathways to the challenging new environmental conditions is taking place is still an open question; even though we know a lot about model organisms like *E. coli*.

With regards to this open questions the present contribution provides new methodology and insights into the dynamic processes taking place during the challenges of nutrient limitations. Not only the analysis of metabolite patterns was undertaken but also selected mutant strain experiments were performed. In particular I like to mention the PTS activity assays using nine different intermediates of the central carbon metabolism to understand the role of pyruvic acid in the adaptation toward environmental stresses. The importance of pyruvate as a candidate metabolite for the regulation of glucose uptake and acetate secretion was well investigated.

An important aspect I see in the clear description of the data without over interpretation and

pronounced speculation. From my point of view in particular the assessment of the role of (sic!) "low cost amino acids" in the bacterial growth is a major outcome of the study.

We thank the reviewer for her/his positive and encouraging comments

The authors used a non-targeted approach to analyze the metabolome. To further investigate the changes in the amino acid pool they used the ACCQ-TAQ assay. Would it be worth to think about a more comprehensive analysis of the extracellular metabolome using methods like NMR? Of course: the limit of detection of NMR is low and compared to MS limited to the database used for the analysis. Nevertheless the opportunities to elucidate further metabolite patterns should be taken into account.

We thank the reviewer for raising this point. In principle one could of course always add more data by different methods. But where would one end? In our particular case, NMR might be able to provide absolute concentration changes on another 10 or maybe 20 compounds, which would not substantially change the data situation where in our case we have relative concentration changes from 427 compounds. The main point of our approach is to make direct use of the broad coverage relative data that untargeted MS can deliver and go straight to hypothesis generation with the new FBA method. These hypotheses are either directly testable (as we did here) or the analysis would allow to conclude which input data are most sensitive to then make a rational choice for more quantitative methods to measure those more accurately. Here, using more quantitative methods such as NMR can indeed add extremely useful inputs for the model. We mention this aspect in the discussion.

The method section is compiled in a very consistent way and provides the necessary data to follow the experimental design by the authors.

The manuscript text needs only some minor spelling check; like "filter sterilized" vs. "filter-sterilized" (page 11, 314, 317). I suppose this could be done in the end review.

OK

Overall the contribution will be of great impact to the community and will certainly influence new approaches to understand the adaptation processes in the bacterial cell towards environmental challenges and the close connection of distinct biochemical pathways.

We sincerely appreciate reviewer's supportive comments.

Reviewer #3 (Remarks to the Author):

In this study, the authors characterize E. coli metabolism in complex medium, by combining measurements of external metabolites (with non-targeted mass-spectrometry) and a constraint-based modeling approach (a kind of dynamical metabolic flux analysis). The model used is the iJO1366 reconstruction of the E. coli metabolism. More precisely, metabolite concentrations are used to determine the secretion and uptake rates along growth. Solving a global optimization problem allows predicting internal metabolic fluxes. Deeper analysis of the predicted fluxes and measured concentrations along growth allows the authors to identify that amino acids are first catabolized, and

then only glucose is consumed while acetate is excreted. Ketoacids like Pyruvate are proposed to regulate the coordination between amino acid and glucose catabolism through inhibition of PtsI activity.

Overall, the paper is interesting for the community, clear and well written. However, it is a pity that the text in the artwork (in the main text and the supplementary information) is often so small that it is impossible to fully interpret the figures when printed. Their screen resolution does not permit zooming too much as well. Table S2 is missing, Table 1 is in fact Table S1, and numbering is shifted at some point in the list of references (e.g ref 31 should be 29).

We thank the reviewer for the positive comments and constructive criticisms. Unfortunately, image resolution was compromised during conversion in the submission. We now extensively revised figures to improve readability. Because of size restriction, Table S2 had to be uploaded by the informatics support of Nat. Comm, and I assume something went wrong in the process. To avoid further confusion, original images and supplementary table can be downloaded also here:

<https://polybox.ethz.ch/index.php/s/IG7WRgKBJaOCoD8>, password: Zampieri

While I found the results convincing in general, I have a number of comments/questions detailed below.

1) Properly estimating external fluxes is crucial in this paper, since most results are derived from them.
a) The authors should provide more details on how they spline fitted the metabolite concentrations and OD600 (beside the small explanation on lines 106-108). They could easily add a section in the supplementary information to describe their approach. Some related concerns follow:

We now clarify the procedure used to fit the experimental data (adaptive regression splines¹⁴) and derive instantaneous uptake/secretion rates by adding full details in the main text. (see also point following point c and d)

b) The phenomena the authors look at happen in the first few hours of growth, when measurements of optical density and concentrations are especially noisy and difficult to interpolate. Spline fitting is known to be particularly sensitive to noise. Uncertainty in the spline fitting propagates to the uptake/secretion/growth rates due to the spline differentiation (and division by fitted OD in the fluxes). Hence, the curve for the growth rate shown in Figure 1 reflects maybe the reality, but it may also (partially) result from fitting problems.

The reviewer has a point here. Indeed, noise in estimates of growth rate (Fig.1A), intracellular fluxes (Fig.2C) and uptake/secretion of glucose and acetate (Fig. 3) during the first growth phase are larger than for later time points, as shown in Fig.1A, Fig.2C and Fig. 3. We are very well aware of this problem and this is why we accounted for measurement noise in the modeling approach by introducing lower and upper bounds to include the variability of experimentally derived exchange fluxes. We clarify this aspect in the main text. The FBA approach is able to further reduce the variability in exchange flux estimates, as shown for glucose uptake and acetate secretion (Fig. 3 red vs grey shaded area).

c) In addition, spline fitting has difficulties to capture sudden changes of concentrations and smooths too much the time-course profiles, which possibly hides transient phenomena. How did the authors deal with this possible problem, given that they are interested in metabolic transitions?

Generally, the extracellular metabolite concentrations do not feature highly transient behavior, unlikely intracellular metabolites. Nevertheless, the Multivariate Adaptive Regression Splines used here is based on piece wise regression models and identify time points (knots) where data follow a different base model. Hence time dependent data can undergo rapid changes in dynamics and the model is able to catch a change in the dynamic trends that are not spurious noise in the measurements. More specifically, MARS models solve the problem of non-parametric fitting with a certain regularization to avoid overfitting. Hence, it automatically determines the number of basis spline functions and knot locations without the necessity for manual curation, yielding estimates robust to noisy outliers¹⁴ (see also point 5). This is made more explicit in the text now, and details can be found in the main description of the FBA methodology and materials and methods.

d) Some fits do not follow well the trend of the data. Are there some constraints in the fitting procedure? See for instance the Proline and Alanine profiles: their negative flux indicates some consumption below OD=1, while the data show a plateau instead.

The fitting procedure was apparently described too shortly. Indeed as noticed by the reviewer for some metabolites, measurements are noisier, like for alanine. We decided to use an automated regularization strategy that doesn't require manual adjustment, like the one proposed in¹⁵ (see also point 5), and that is robust to outliers. To further take into account the potential effects of outliers in the fitting estimates, we used a bootstrapping approach, which excludes 10% of all data-points and uses the same adaptive regression spline to fit the data. The resulting estimate is the average +/- standard deviation across the sampled data points. Hence, some of the trends that are apparent on a first eye examination, might not be robust to the random down-sampling of the data. Overall, we found a very good agreement with measurements of amino acids abundance and fitted values. We clarify this aspect in the description of the FBA method.

2) Section "Dynamic coordination of amino acid and glucose consumption"

a) lines 181-184: The authors conclude from the amino acid supplementations shown in Figure S11 that the uptake rates do not depend on the amino acid concentration. This is certainly true for Asp, Ser and Gly that are consumed before Glucose, even at 0.25% CAA. However, this is not always true for other amino acids that seem to be co-consumed with Glucose at low CAA concentration, while they are consumed after glucose depletion at higher CAA concentration. See for instance the case of Trp.

This is an important aspect that we did apparently not explain sufficiently. We expanded this section, included new analysis and experiments to address the observations raised by the reviewer. Average consumption of amino acids strongly correlate among conditions. Hence, whether amino acids are fully consumed before or after glucose is depleted, largely depend on the initial concentration of the amino acid. Yet the reviewer has a point and some amino acids exhibit inconsistent results. We now performed new experiments and analysis to investigate this aspect more in details

In this study, we found that the average amount of amino acid consumed per unit change in OD (i.e. initial amino acid concentration divided by the OD at time of depletion) is inversely related to the amino acid metabolic cost, defined as the number of high-energy phosphate bonds required for biosynthesis¹⁶. Our data hence support the hypothesis that amino acids biosynthetic cost imposed a selective pressure to encode less costly amino acids in highly abundant protein¹⁷ (Figure S11 third column). This relationship can be observed also when comparing biosynthesis cost to an average estimate of amino

acids uptake rates, calculated as the initial amino acid concentration divided by the hours and culture gram of dry biomass (gDW) at time of amino acid depletion (see new Fig. 4C). We now performed new experiments to include 3 more conditions, where not only the absolute abundance of amino acid was changed but also their relative abundance to each other. We supplemented the medium with 3 equimolar concentrations of all 20 amino acids supplemented. We show that these estimates largely correlate across different amino acid concentrations supplemented to the medium (new Fig. S12). In particular, average consumption of Tryptophan (Trp) is relatively constant across different CAA concentrations (new Fig. 4D).

However, as suggested by the reviewer for some amino acids (e.g. aspartate, glutamate) we observed larger fluctuations in their uptake rates depending on the overall composition of amino acids in the tested mixes. This observation suggests for additional regulatory mechanisms relating amino acid uptake rates between each other. To test this hypothesis, we calculated pairwise correlation between amino acid uptake rates across the 7 tested conditions (new Fig. 4 E) and found several strong positive and negative correlations between amino acids uptake rates. These correlations suggest for some degree of dependency in the uptake rates of amino acids.

The strongest correlation was found between valine and iso-/leucine. Notably, at CAA concentrations lower than or equal to 0.5g/L we noticed a remarkable drop in growth rate (coinciding with depletion of amino acids like Iso-/Leucine and asparagine) (see Fig. S11 first column), followed by its resumption after valine depletion. We hence performed new experiments showing that by supplementing M9 glucose+ 0.25g/L CAA, with leucine or isoleucine we can abolish the sudden slow down in growth (see Fig. S13), validating the functional interaction between valine and iso-/leucine consumption.

b) lines 185-189: Why should the cost of amino acid synthesis drive mostly the order in which amino acids are catabolized? The authors do not fully interpret the supplementary results in Figure S11 (notably the two last columns), but this could be helpful to follow their line of reasoning. What is their interpretation? c) Same lines. An alternative interpretation to the synthesis cost could be the following one: according to the third column (and if I can read correctly due to the low figure resolution), it seems that amino acids that are consumed first are also among the most needed to generate new cells (see Ser for instance). Therefore, the ordering in the consumption of amino acids and glucose could simply reflect a cell demand. In the same line, and if I understood correctly the y-axis in the last column (how exactly are determined the average relative amino acid concentrations among multiple environments?), these amino acids are a bit less abundant than in many environments, which puts a pressure on cells to import these needed amino acids and to catabolize them in order to grow and divide.

We did not find significant differences in the order in which the amino acids are consumed (see also point above). Rather, we show that while most amino acids are immediately taken up, they are metabolized differently. The ones with lower biosynthetic cost are catabolized. More specifically, in Fig. S11 we compared the average amount of amino acid consumed per unit change in OD (which strongly correlate with average uptake rates – Fig. 4C) with (i) biosynthetic cost, (ii) contribution in biomass generation and (iii) their abundance across different environments (we collected supplementary data reported in ¹⁸).

We cannot conclude which of these factors is responsible for the differences in average uptake rates of amino acids. We clarify this point better in the text now. However, as correctly noticed by the reviewer, the current working hypothesis is that bacteria have evolved to optimize usage of proteins, i.e. highly

abundant proteins are more likely to be composed of cheap amino acids¹⁷. It is plausible that because amino acids like methionine or tryptophan are so costly to produce and lowly abundant in the environment, bacteria don't degrade them to avoid possible waste of resources. In fact we show that E.coli uses a very similar consumption/utilization program, regardless of the extracellular abundance of amino acids: it rapidly consumes cheap amino acids, which often can be degraded into carbon or nitrogen sources (i.e. serine), and matches consumption rate of costly amino acids to their requirement for protein synthesis. We now clarify the interpretation of the correlative signatures between average uptake rates and amino acids costs.

3) Section "The role of pyruvate in coordinating glucose catabolism"

The authors monitored Crp activity but could not involve this factor in the reduced glucose consumption. Did the authors also look for changes in PdhR activity in response to amino acid addition?

In the first submission we did not assess PdhR activity, but in response to the raised point we now assess the potential role of PdhR during growth with amino acids by measuring growth rate of Δ pdhR in M9 glucose and M9 glucose + CAA (Figure S19). We found that Δ pdhR exhibits mild but significant growth reduction only in M9 glucose, suggesting that upon addition of CAA, the residual activity of PDHR is completely repressed by increased pyruvate levels (as expected). However, previous flux data¹⁹ shows that PdhR deletion doesn't affect glucose or galactose uptake-rates. Overall, regulation of PdhR activity by pyruvate can further contribute in conferring to pyruvate a key role in the coordination of carbon and amino acid catabolism.

In addition, to address comments from reviewer 1 and to better understand the role of transcriptional regulation in mediating the consumption and utilization of amino acids, we monitored Crp activity upon addition of amino acids in M9 succinate minimal medium. We selected succinate because it is not transported by the PTS system and is known to induce high levels of Crp activity (new Fig.5 A). We found that similarly to glucose, addition of CAA strongly reduces succinate uptake. However, differently from glucose, pyruvate levels are not affected while Crp activity correlates with succinate uptake rate (Fig. 5A and 6C). These additional experimental evidence suggest that while consumption of PTS carbon source and amino acids can be regulated at a post-translational level, consumption of non-PTS carbon sources, like succinate, and amino acids are likely regulated mainly transcriptionally via Crp.

4) Section Methods

a) Line 426: why do the authors impose the growth rate as a hard constraint? This could be problematic if there is some uncertainty on the determination of the growth rate (e.g. for the reasons given in comment 1b). Why not trying to minimize the L1-norm distance between the predicted and measured growth rate, along with the uptake and secretion rates?

Differently from the metabolomics measurements, OD measurements were taken at a much higher time resolution (10 min using a plate reader). This indeed offer the possibility to catch fast changes in growth rates and improve the ability of the model to interpret such rapid changes. Moreover, growth rates and uptake rates are typically expressed in different units (1/hours vs mmol/g of dry biomass/h). These differences can bias the fitting, i.e. differently penalize deviation of model predictions from growth rate vs uptake/secretion rates. Hence we prefer to avoid these problems by imposing a hard constraint on biomass production.

b) Is the computer code available to allow reproducing the results?

The matlab code to reproduce the results will be available upon publication on the group website (<http://www.imsb.ethz.ch/research/zampieri-group.html>) and in the meantime here: <https://polybox.ethz.ch/index.php/s/lG7WRgKBJaOCOD8>, password: Zampieri

5) Additional comments/questions:

a) The uptake and excretion of metabolites is correlated with the accumulation of biomass. This information could be used in the data spline fitting, instead of fitting the curves separately (see doi: 10.1093/bioinformatics/btx250 for an example). This could constrain the fits more and possibly give more accurate estimations of the rates.

Thanks for this insightful comment and suggestion. We were not aware of the article suggested by the reviewer. The approach described in ¹⁵ is an elegant alternative way to directly couple metabolite and biomass (OD) measurements using a dynamical smoothing approach based on Kalman filtering. We imputed our data in the rate estimation algorithm suggested by the reviewer (<https://team.inria.fr/ibis/rate-estimation-software/>), hoping to compare results of this analysis with our estimates. However, we had hard time to get stable convergence and reasonable estimates, possibly because of the large number of rates to be estimated. If we understand correctly, the algorithm needs a manual curation of the results, and selection of metabolic-switch points. In ¹⁵ the authors describe this approach using the example of E. coli undergoing diauxic shift during growth in minimal glucose medium. This manual curation is less intuitive when growth is monitored in complex medium with continuous depletion of nutrients and changes in growth rate. Hence, we avoid to make any inconclusive comparison and we explicitly mention that other strategies can be used to process measurements of metabolite abundance in the supernatant.

b) What is the need to select the solution minimizing the sum of absolute fluxes in the global optimization problem? The minimization of the distance between measured and predicted external fluxes together with the constraints on fluxes do not suffice to reduce the space of solutions to solutions with physiologically reasonable values?

Minimization of the total sum of fluxes is widely used to minimize futile cycles and is here adopted to select for the most biologically plausible flux solutions. In particular we want to avoid scenarios in which the model overestimates the uptake rate of a nutrient that could be compensated by a corresponding overestimate of secretion of close-by byproducts.

6) Minor comments and misspellings:

a) Methods: line 428: Di is not defined

OK

b) Figure 1: the shaded region in panel A is not described in the legend. Misspellings in panel B: deteceted->detected

OK

c) Figure 5, panel C: akg should be displayed in green in the figure legend.

OK

d) Table 1: add a column with SD on the Acetate/Glucose ratios.

OK

e) Figure S1, right panel: display the SD for the two growth rates.

f) Figure S11: mention that y-axes in the fourth column are on a log scale. Provide more information in legend on how the average relative amino acid concentrations across multiple environments are determined.

OK

g) Same figure, labels A to D are missing.

OK

h) Figure S15, title: glucose->glucose; axis label: Maximu->Maximum

OK

i) Main text, line 177: Figure 3, dashed green line->green line

OK

j) Main text, line 432: in in gram->in gram

OK

k) Main text, bibliography: reference style is heterogeneous.

OK

l) Artwork: improve resolution and text size in Fig 1B, 3, S10, S11

OK

Bibliography

1. Lewis, N. E. *et al.* Omic data from evolved E. coli are consistent with computed optimal growth from genome-scale models. *Mol Syst Biol* **6**, (2010).

2. Campos-Bermudez, V. A., Bologna, F. P., Andreo, C. S. & Drincovich, M. F. Functional dissection of Escherichia coli phosphotransacetylase structural domains and analysis of key compounds involved in activity regulation. *FEBS J.* **277**, 1957–1966 (2010).
3. Albe, K. R., Butler, M. H. & Wright, B. E. Cellular concentrations of enzymes and their substrates. *J. Theor. Biol.* **143**, 163–195 (1990).
4. Zimmermann, M., Sauer, U. & Zamboni, N. Quantification and Mass Isotopomer Profiling of α -Keto Acids in Central Carbon Metabolism. *Anal. Chem.* **86**, 3232–3237 (2014).
5. Venditti, V., Ghirlando, R. & Clore, G. M. Structural Basis for Enzyme I Inhibition by α -Ketoglutarate. *ACS Chem. Biol.* **8**, 1232–1240 (2013).
6. Venditti, V., Tugarinov, V., Schwieters, C. D., Grishaev, A. & Clore, G. M. Large interdomain rearrangement triggered by suppression of micro- to millisecond dynamics in bacterial Enzyme I. *Nat. Commun.* **6**, 5960 (2015).
7. Doucette, C. D., Schwab, D. J., Wingreen, N. S. & Rabinowitz, J. D. α -Ketoglutarate coordinates carbon and nitrogen utilization via enzyme I inhibition. *Nat. Chem. Biol.* **7**, 894–901 (2011).
8. Piazza, I. *et al.* A Map of Protein-Metabolite Interactions Reveals Principles of Chemical Communication. *Cell* **172**, 358-372.e23 (2018).

9. Link, H., Fuhrer, T., Gerosa, L., Zamboni, N. & Sauer, U. Real-time metabolome profiling of the metabolic switch between starvation and growth. *Nat. Methods* **12**, 1091–1097 (2015).
10. Li, X., Gianoulis, T. A., Yip, K. Y., Gerstein, M. & Snyder, M. Extensive In Vivo Metabolite-Protein Interactions Revealed by Large-Scale Systematic Analyses. *Cell* **143**, 639–650 (2010).
11. Hegewald, E. & Knorre, W. A. Kinetics of growth and substrate consumption of *Escherichia coli* ML 30 on two carbon sources. *Z. Für Allg. Mikrobiol.* **18**, 415–426 (1978).
12. Morgan, M. J. & Kornberg, H. L. Regulation of sugar accumulation by *Escherichia coli*. *FEBS Lett.* **3**, 53–56 (1969).
13. Rühl, M., Le Coq, D., Aymerich, S. & Sauer, U. ¹³C-flux analysis reveals NADPH-balancing transhydrogenation cycles in stationary phase of nitrogen-starving *Bacillus subtilis*. *J. Biol. Chem.* **287**, 27959–27970 (2012).
14. Friedman, J. H. Multivariate Adaptive Regression Splines. *Ann. Stat.* **19**, 1–67 (1991).
15. Cinquemani, E., Laroute, V., Coccagn-Bousquet, M., de Jong, H. & Ropers, D. Estimation of time-varying growth, uptake and excretion rates from dynamic metabolomics data. *Bioinforma. Oxf. Engl.* **33**, i301–i310 (2017).

16. Akashi, H. & Gojobori, T. Metabolic efficiency and amino acid composition in the proteomes of *Escherichia coli* and *Bacillus subtilis*. *Proc. Natl. Acad. Sci.* **99**, 3695–3700 (2002).
17. Mee, M. T., Collins, J. J., Church, G. M. & Wang, H. H. Syntrophic exchange in synthetic microbial communities. *Proc. Natl. Acad. Sci.* **111**, E2149–E2156 (2014).
18. Moura, A., Savageau, M. A. & Alves, R. Relative Amino Acid Composition Signatures of Organisms and Environments. *PLoS ONE* **8**, e77319 (2013).
19. Haverkorn van Rijsewijk, B. R. B., Nanchen, A., Nallet, S., Kleijn, R. J. & Sauer, U. Large-scale ¹³C-flux analysis reveals distinct transcriptional control of respiratory and fermentative metabolism in *Escherichia coli*. *Mol Syst Biol* **7**, (2011).

REVIEWERS' COMMENTS:

Reviewer #1 (Remarks to the Author):

Authors have answered most of the points in a very clear way. However, there are still some questions regarding the way the method is described in the revised text. For example there is some overlap between the Lines 121 and 501. Also I could not see the following points in the methods section although authors indicate they have added them to the paper.

1)" We emphasize that the metabolites for which we measured absolute concentrations in the supernatant are glucose, acetate and 20 amino acids. For ~400 metabolites we measured relative levels in the supernatant. " This statement should be somewhere explicit in the paper and in the methods section.

2) "These objectives are optimized one after the other, so no weights are used. First, we restrict the space of scaling parameters c and flux solutions to those minimizing the distance between predicted and measured uptake rates of glucose and amino acids, secretion of acetate and growth rate. Next this objective becomes a new constraint in the following optimization, where we minimize the distance between uptake-secretion rates of all metabolites measured only in relative amount. Finally, the second objective becomes a new additional constraint in the model. To restrict the space to the most meaningful flux solutions (i.e. avoid solutions with futile cycles), we used a classical minimization of the total sum of absolute fluxes"

Also, I think the authors should formulate the exact sequence of optimization problems as part of the SI and refer to this in the methods. For example it is important to understand exactly how the constraints from the first optimization appear in the second and the final optimization problem. The way it is written in the methods section there are three different objective functions. Either include the final formulation with the first two optimization problems as constraints or include the initial optimization problem and refer to the SI for the final optimization problem. Either way the formulation for both the final, the initial and intermediate steps need to be clearly described.

3. "To more specifically address reviewer's comment on Fig. 2 (same apply to Fig. 3) results from flux variability analysis (i.e. how noise in the measurements propagate to flux estimation) are highlighted by the grey dark zones. Hence our final estimates of fluxes already account for noise $\tilde{v}(t)$ are within the exometabolome derived average \pm standard deviation of estimated absolute and relative exchange fluxes measurements (reported in Table S1), and we now more carefully described estimates of measurement variability and how noise in the measurements are considered in the modeling"

This is great. Thank you very much. Please indicate in your revised methods section the way you calculated to the confidence intervals or the grey shaded areas.

Reviewer #3 (Remarks to the Author):

The authors carefully addressed all the points made previously. The new results and analyses are convincing. I think the paper has improved significantly and recommend it for publication.

REVIEWERS' COMMENTS:

Reviewer #1 (Remarks to the Author):

Authors have answered most of the points in a very clear way.

We thank the reviewer for the constructive and positive comments.

However, there are still some questions regarding the way the method is described in the revised text. For example there is some overlap between the Lines 121 and 501.

We revised the text to avoid repetition

Also I could not see the following points in the methods section although authors indicate they have added them to the paper.

1)" We emphasize that the metabolites for which we measured absolute concentrations in the supernatant are glucose, acetate and 20 amino acids. For ~400 metabolites we measured relative levels in the supernatant. " This statement should be somewhere explicit in the paper and in the methods section.

We now clarify this point in second paragraph of the results section.

2) "These objectives are optimized one after the other, so no weights are used. First, we restrict the space of scaling parameters c and flux solutions to those minimizing the distance between predicted and measured uptake rates of glucose and amino acids, secretion of acetate and growth rate. Next this objective becomes a new constraint in the following optimization, where we minimize the distance between uptake-secretion rates of all metabolites measured only in relative amount. Finally, the second objective becomes a new additional constraint in the model. To restrict the space to the most meaningful flux solutions (i.e. avoid solutions with futile cycles), we used a classical minimization of the total sum of absolute fluxes"

Also, I think the authors should formulate the exact sequence of optimization problems as part of the SI and refer to this in the methods. For example it is important to understand exactly how the constraints from the first optimization appear in the second and the final optimization problem. The way it is written in the methods section there are three different objective functions. Either include the final formulation with the first two optimization problems as constraints or include the initial optimization problem and refer to the SI for the final optimization problem. Either way the formulation for both the final, the initial and intermediate steps need to be clearly described.

We revised the methods section, and now formulate the problem with the first two objectives as constraints. We clarify the procedure in the corresponding method section.

3. "To more specifically address reviewer's comment on Fig. 2 (same apply to Fig. 3) results from flux variability analysis (i.e. how noise in the measurements propagate to flux estimation) are highlighted by the grey dark zones. Hence our final estimates of fluxes already account for noise $\tilde{v}(t)$ are within the

exometabolome derived average \pm standard deviation of estimated absolute and relative exchange fluxes measurements (reported in Table S1), and we now more carefully described estimates of measurement variability and how noise in the measurements are considered in the modeling"

This is great. Thank you very much. Please indicate in your revised methods section the way you calculated to the confidence intervals or the grey shaded areas.

We now clarify this point and emphasize the corresponding section in the methods section

Reviewer #3 (Remarks to the Author):

The authors carefully addressed all the points made previously. The new results and analyses are convincing. I think the paper has improved significantly and recommend it for publication.

We thank the reviewer for his/her criticisms and comments which help us to mature the manuscript